# Investigating the Impact of Model Width and Density on Generalization in Presence of Label Noise

Yihao Xue[1]          Kyle Whitecross[2]          Baharan Mirzasoleiman[1]

[1]Computer Science Department, University of California
[2]College of Information and Computer Sciences, University of Massachusetts Amherst

## Abstract

Increasing the size of overparameterized neural networks has been a key in achieving state-of-the-art performance. This is captured by the double descent phenomenon, where the test loss follows a decreasing-increasing-decreasing pattern (or sometimes monotonically decreasing) as model width increases. However, the effect of label noise on the test loss curve has not been fully explored. In this work, we uncover an intriguing phenomenon where label noise leads to a *final ascent* in the originally observed double descent curve. Specifically, under a sufficiently large noise-to-sample-size ratio, optimal generalization is achieved at intermediate widths. Through theoretical analysis, we attribute this phenomenon to the shape transition of test loss variance induced by label noise. Furthermore, we extend the final ascent phenomenon to model density and provide the first theoretical characterization showing that reducing density by randomly dropping trainable parameters improves generalization under label noise. We also thoroughly examine the roles of regularization and sample size. Surprisingly, we find that larger $\ell_2$ regularization and robust learning methods against label noise exacerbate the final ascent. We confirm the validity of our findings through extensive experiments on ReLu networks trained on MNIST, ResNets/ViTs trained on CIFAR-10/100, and InceptionResNet-v2 trained on Stanford Cars with real-world noisy labels.

## 1 INTRODUCTION

Training neural networks of ever-increasing size on large datasets has played a pivotal role in achieving state-of-the-art generalization performance across various tasks [Devlin et al., 2018, Dosovitskiy et al., 2020, Brown et al., 2020, Radford et al., 2021]. While large unlabeled data can often be easily collected, obtaining high-quality labels for these datasets is prohibitively expensive, leading to the use of labeling techniques, such as crowd-sourcing and automatic labeling, that introduce a significant amount of label noise [Krishna et al., 2016]. Unfortunately, the impact of label noise on the generalization behavior of neural networks in relation to model size has not been comprehensively examined.

Recent research has aimed to reconcile the generalization benefits of increasing the size of overparameterized neural networks with the classical bias-variance trade-off that advocates for intermediate model size. To this end, the double descent phenomenon [Belkin et al., 2019, Spigler et al., 2019] has been proposed, suggesting that the test loss initially follows a U-shape but then descends again once the model is overparameterized. At times, the U-shaped behavior may not be evident, and the curve may seem to be monotonically decreasing[Nakkiran et al., 2021]. The works of [Yang et al., 2020, Adlam and Pennington, 2020a] attribute the double descent to the combination of decreasing bias and the unimodal *variance* of the test loss w.r.t. model width. However, the effect of label noise on the double descent phenomenon has not been fully explored.

In this work, we reveal that label noise can significantly alter the shape of the loss curve. Our study uncovers an intriguing phenomenon, which we refer to as the *final ascent*, manifesting either a *U-shape or a decreasing-increasing-decreasing-increasing* pattern. In a wide range of settings, label noise leads to an eventual ascent in the original loss curve, distinguishing it from both the double descent or monotonically decreasing curve. Through theoretical analysis, we show that this occurs because sufficiently large *noise-to-sample size ratio* transforms the variance of the test loss from unimodal to an increasing-decreasing-increasing shape. We also provide further insights into the final ascent. Firstly, perhaps surprisingly, applying stronger $\ell_2$ regularization only exacerbates the phenomenon as it enables intermediate model sizes to achieve lower test loss. Secondly, increasing

the sample size can alleviate the final ascent.

Furthermore, we add *model density*, the fraction of weights that are trainable, as a new dimension to the discussion on generalization. We provide the first theoretical characterization of final ascent w.r.t. model density, showing that optimal generalization occurs at intermediate density levels under label noise. We also make two significant findings. Firstly, as density decreases, the optimal width increases and achieves lower test loss, highlighting the advantage of wider but sparser models. Secondly, while reducing density has a similar effect to stronger $\ell_2$ regularization in theory, in practice, adjusting the density of neural networks achieves even lower density compared to adjusting the regularization.

We also empirically examine the final ascent phenomenon when models are trained with robust learning algorithms that are designed to counteract the effect of label noise. Our results demonstrate that, similar to the impact of $\ell_2$ regularization, SOTA robust algorithms [Liu et al., 2020, Li et al., 2020] amplify the final ascent phenomenon. Notably, reducing model density can further improve the SOTA performance for robust training methods, such as DivideMix [Li et al., 2020] and ELR [Liu et al., 2020], as shown in Figures 30 and 32. This suggests that models with intermediate width or density can be even more beneficial when these algorithms are used, which is typically the case in practical scenarios.

Our findings are supported by extensive experiments. We confirm the validity of our results across various settings, including training two-layer networks on MNIST [LeCun, 1998], ResNets [He et al., 2016] and ViT [Dosovitskiy et al., 2020] on CIFAR-10 and CIFAR-100 [Krizhevsky et al., 2009], and InceptionResNet-v2 [Szegedy et al., 2017] on Stanford Cars with real label noise[Jiang et al., 2020]. We provide an in-depth discussion of how different factors affect the final ascent. Notably, the final ascent can even occur with only 20% label noise.

Our results highlight the following important messages:

- It is important to use large models with care in presence of label noise,
- Wider but sparser models contribute to improved generalization under label noise,
- Above considerations are of increased significance when data is limited, or when strong regularization or robust methods are used.

## 2 RELATED WORK

**Double descent.** Contrary to the classical learning theory that advocates for intermediate model size, recent works have shown that increasing the size of overparameterized neural networks only improves generalization [Neyshabur et al., 2014]. This is explained by the double descent phenomenon [Belkin et al., 2019, Spigler et al., 2019], which

posits that the test loss initially follows a U-shape and then descends again once the model is overparameterized. This phenomenon has been theoretically investigated in various regression settings, including one-layer linear [Belkin et al., 2020, Derezinski et al., 2020, Kuzborskij et al., 2021], random feature [Hastie et al., 2019, Mei and Montanari, 2019, Adlam and Pennington, 2020a, Yang et al., 2020, d'Ascoli et al., 2020], and NTK [Adlam and Pennington, 2020b] regressions. While the primary focus has been on the behavior of the test loss [Hastie et al., 2019, Belkin et al., 2020, Derezinski et al., 2020, Adlam and Pennington, 2020b], some works have decomposed it into bias and variance [Mei and Montanari, 2019, Yang et al., 2020], and a few have further decomposed the variance into multiple sources, including label noise [Adlam and Pennington, 2020a, d'Ascoli et al., 2020]. However, prior studies primarily focused on how label noise intensifies the double descent curve's peak [Adlam and Pennington, 2020a, Nakkiran et al., 2021]. We discover the novel concept of *final ascent*, for the first time, and show that increased regularization—via $\ell_2$ or robust methods—that is essential to train robustly in presence of label noise, exacerbates the final ascent. This makes our study distinct from prior work [Adlam and Pennington, 2020a, Nakkiran et al., 2021, d'Ascoli et al., 2020], which did not account for the role of regularization/robust methods, and hence failed to observe the final ascent. [Nakkiran et al., 2020] considered regularization, but explored a different phenomenon than us, showing that optimal regularization tuned for each width can eliminate the peak in a double descent curve. In summary, our finding challenges the prevailing view that larger models always perform better, showing that increasing model size can have detrimental effects under noise and regularization. Moreover, our study is the first to assess model density's effect, showing its potential to achieve SOTA with robust methods (Fig 30 31 32).

**Neural Network Density.** [Frankle and Carbin, 2018, Lee et al., 2018, Frankle et al., 2019, Wang et al., 2020, Frankle et al., 2020, Tanaka et al., 2020] proposed methods to reduce model density (pruning) for improved training and inference efficiency. [Jin et al., 2022] empirically investigated the impact of density on generalization using a pruning technique [Renda et al., 2020] involving training and rewinding. However, the effect of density as a factor of *model size* has remained poorly understood, as pruning techniques incorporate additional information from training, initialization, or gradients. In contrast, we randomly drop weights thus isolating the effect of model size from other factors. We provide theoretical analysis of random feature regression, complementing the empirical nature of the aforementioned works. Exploring the effects of specific pruning techniques on the final ascent phenomenon uncovered in this paper is an interesting future work.

**Robust Methods.** Extensive efforts have been made to develop methods for robust learning against noisy labels

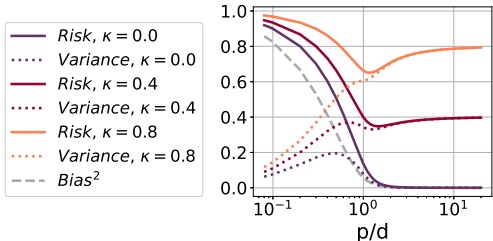

Figure 1: Decomposition of test loss. **Risk = Bias² + Variance**. **Bias²** always monotonically decreases. **Variance** exhibits a transition from a unimodal shape to an increasing-decreasing-increasing pattern as noise increases, leading to the final ascent in test loss.

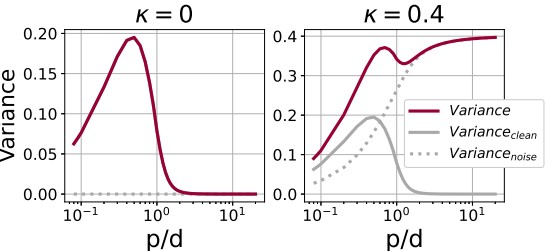

Figure 2: Decomposition of variance. **Variance = Variance**clean **+ Variance**noise. **Variance**clean is always unimodal. **Variance**noise monotonically increases with width, and its scale grows with noise level, leading to the increasing-decreasing-increasing pattern of **Variance** at sufficient noise.

[Zhang and Sabuncu, 2018, Jiang et al., 2018, Han et al., 2018, Mirzasoleiman et al., 2020, Liu et al., 2020, Li et al., 2020, Xia et al., 2019]. These methods serve as a regularization to mitigate label noise. We will demonstrate that robust methods exacerbate the final ascent, i.e., reducing the model width or density can yield even greater benefits when employing robust methods.

## 3  THEORETICAL ANALYSIS OF RANDOM FEATURE RIDGE REGRESSION

We conduct a theoretical analysis of label noise's effect on the test loss curve in a linear neural network with a random first layer. Random feature regression has been the go-to model in studying the double descent phenomenon [Hastie et al., 2019, Mei and Montanari, 2019, Yang et al., 2020, Adlam and Pennington, 2020a, d'Ascoli et al., 2020], owing to its theoretical tractability. We note that studying regression tasks is a widely adopted approach for understanding neural networks' generalization behavior [Hastie et al., 2019, Mei and Montanari, 2019, Advani et al., 2020, Bartlett et al., 2020] and yields meaningful conclusions that extend to *classification* tasks, as is confirmed by prior work Yang et al. [2020], d'Ascoli et al. [2020], Nakkiran et al. [2020] and we will also show experimentally.

We start by showing that sufficiently large noise-to-sample-size ratio introduces a final ascent to the loss curve, and explain it through the shape of variance. Then, we provide the first theoretical study on the effect of model density, showing the benefit of reducing density. These findings will be validated in various neural network classification tasks in Sec 4.

### 3.1  EFFECT OF WIDTH: THE FINAL ASCENT

Suppose we have a training dataset $(\boldsymbol{X}, \boldsymbol{y})$ where $\boldsymbol{X} = [\boldsymbol{x}_1, \boldsymbol{x}_2, \ldots, \boldsymbol{x}_n]$ and $\boldsymbol{y} = [y_1, y_2, \ldots, y_n]^\top$. Each input $\boldsymbol{x}_i \in \mathbb{R}^d$ is independently drawn from a Gaussian distribution $\mathcal{N}(0, \mathbf{I}_d/d)$. Each label $y_i \in \mathbb{R}$ is generated as $y_i = \boldsymbol{x}_i^\top \boldsymbol{\theta} + \epsilon_i$. Here, $\boldsymbol{\theta} \in \mathbb{R}^d$ has entries independently

drawn from $\mathcal{N}(0, 1)$, and $\epsilon_i \in \mathbb{R}$ is the label noise drawn from $\mathcal{N}(0, \sigma^2)$ for each $\boldsymbol{x}_i$. We learn a two-layer linear network where the first layer $\boldsymbol{W} \in \mathbb{R}^{p \times d}$ has entries randomly drawn from $\mathcal{N}(0, 1/d)$ and the second layer is given by

$$
\begin{aligned}
\hat{\boldsymbol{\beta}} &= \arg\min_{\boldsymbol{\beta} \in \mathbb{R}^p} \|(\boldsymbol{W}\boldsymbol{X})^\top \boldsymbol{\beta} - \boldsymbol{y}\|^2 + \lambda \|\boldsymbol{\beta}\|^2 \\
&= (\boldsymbol{W}\boldsymbol{X}\boldsymbol{X}^\top\boldsymbol{W}^\top + \lambda \mathbf{I})^{-1} \boldsymbol{W}\boldsymbol{X}(\boldsymbol{X}^\top\boldsymbol{\theta} + \boldsymbol{\epsilon}),
\end{aligned}
$$

where $\boldsymbol{\epsilon} = [\epsilon_1, \epsilon_2, \ldots, \epsilon_n]^\top$. Given a test example with non-noisy label $(\boldsymbol{x}, y)$ where $\boldsymbol{x} \sim \mathcal{N}(0, \mathbf{I}_d/d)$ and $y = \boldsymbol{x}^\top\boldsymbol{\theta}$, the prediction of the learned model is given by $f(\boldsymbol{x}) = (\boldsymbol{W}\boldsymbol{x})^\top \hat{\boldsymbol{\beta}}$. The expected risk (test loss) can be written as

$$
\begin{aligned}
\textbf{Risk} &= \mathbb{E}_{\boldsymbol{\theta}} \mathbb{E}_{\boldsymbol{x}} \mathbb{E}_{\boldsymbol{X}, \boldsymbol{W}, \boldsymbol{\epsilon}} (f(\boldsymbol{x}) - y)^2 \quad (1) \\
&= \underbrace{\mathbb{E}_{\boldsymbol{\theta}} \mathbb{E}_{\boldsymbol{x}} (\mathbb{E}_{\boldsymbol{X}, \boldsymbol{W}, \boldsymbol{\epsilon}} f(\boldsymbol{x}) - y)^2}_{\textbf{Bias}^2} + \underbrace{\mathbb{E}_{\boldsymbol{\theta}} \mathbb{E}_{\boldsymbol{x}} \mathbb{V}_{\boldsymbol{X}, \boldsymbol{W}, \boldsymbol{\epsilon}} f(\boldsymbol{x})}_{\textbf{Variance}} \\
&= \underbrace{\frac{1}{d} \|\mathbb{E}_{\boldsymbol{X}, \boldsymbol{W}} \boldsymbol{B} - \mathbf{I}\|_F^2}_{\textbf{Bias}^2} + \underbrace{\frac{1}{d} \mathbb{E}_{\boldsymbol{X}, \boldsymbol{W}} \|\boldsymbol{B} - \mathbb{E}_{\boldsymbol{X}, \boldsymbol{W}} \boldsymbol{B}\|_F^2}_{\textbf{Variance}_{\text{clean}}} \\
&\quad + \underbrace{\frac{\sigma^2}{d} \mathbb{E}_{\boldsymbol{X}, \boldsymbol{W}} \|\boldsymbol{A}\|_F^2}_{\textbf{Variance}_{\text{noise}}}, \quad (2)
\end{aligned}
$$

where $\boldsymbol{A} := \boldsymbol{W}^\top (\boldsymbol{W}\boldsymbol{X}\boldsymbol{X}^\top\boldsymbol{W}^\top + \lambda\mathbf{I})^{-1}\boldsymbol{W}\boldsymbol{X}$ and $\boldsymbol{B} := \boldsymbol{A}\boldsymbol{X}^\top$ (see the derivation in Appendix A.1). Eq. 1 decomposes the risk into bias and variance, and Eq. (2) subsequently breaks down the variance into two terms. The second term, **Variance**noise, captures the impact of label noise.

Our analysis is conducted under the high-dimensional asymptotic limit where $n$, $d$, and $p$ tend to infinity, while maintaining the ratios $\frac{n}{d} = \psi$, $\frac{p}{d} = \gamma$, and $\frac{\sigma^2}{(n/d)} = \kappa$ constant. To simplify the analysis further, we set $\psi = \infty$. This limit is consistent with the one used in [Yang et al., 2020], which has been shown to capture important features of the double descent. Our numerical experiments in Sec 4.1 confirms that our conclusions hold in a broader range of settings outside of this regime. We note that the noise-to-sample size ratio $\kappa$ remains finite, despite the infinite value of $\psi$. The explicit expression of **Variance**noise is shown below.

**Theorem 3.1.** *For a 2-layer linear network with $p$ hidden neurons and a random first layer, consider learning the second layer by ridge regression with regularizer $\lambda$ on $n$ training examples with feature dimension $d$, and label noise with variance $\sigma$. Let $\lambda = \frac{n}{d}\lambda_0$ and $\sigma^2 = \frac{n}{d}\kappa$ for some fixed $\lambda_0$ and $\kappa$. The asymptotic expression (where $n, d, p \to \infty$ with $\frac{n}{d} = \infty$ and $\frac{p}{d} = \gamma$) of **Variance**$_{noise}$ is given by*

$$\frac{\kappa}{2}\left(\gamma + 2\lambda_0 + 1 - \frac{\gamma^2 + (3\lambda_0 - 2)\gamma + 2\lambda_0^2 + 3\lambda_0 + 1}{\sqrt{\gamma^2 + (2\lambda_0 - 2)\gamma + \lambda_0^2 + 2\lambda_0 + 1}}\right)$$

We provide the derivation of **Variance**$_{noise}$ in Appendix A.3. Note that **Bias**$^2$ monotonically decreases and **Variance**$_{clean}$ is unimodal [Yang et al., 2020] (see Appendix A.2 for expressions of **Bias**$^2$ and **Variance**$_{clean}$). By plotting the theoretical expressions, we make the following key observations.

**The Noise-dependent Variance Shapes the Final Ascent.** Fig 2 shows **Variance** by solid purple, **Variance**$_{clean}$ in solid gray, and **Variance**$_{noise}$ in dashed gray. We see that **Variance**$_{clean}$ is unimodal, while **Variance**$_{noise}$ monotonically increases with width. Thm 3.1 tells that **Variance**$_{noise}$ scales with noise-to-sample size ratio ($\kappa$). Intuitively, variance measures the sensitivity to fluctuations in data and training. As the noise level increases, it has the potential to introduce greater inconsistency among the outputs of models, and this inconsistency grows as the model size increases. We see that as $\kappa$ increases, **Variance**$_{noise}$ becomes more pronounced in the total variance, resulting in an *increasing-decreasing-increasing* trend. Therefore, the risk, which is the sum of **Variance** and the monotonically increasing **Bias**$^2$ (Fig 1), finds its minimum at an intermediate width under sufficiently large $\kappa$.

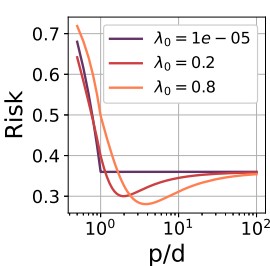

**Regularization Exacerbates Final Ascent.** Regularization is often used to improve robustness to noise, leading to the expectation that it would mitigate the impact of label noise and alleviate the final ascent. However, Fig.3 shows that stronger $\ell_2$ regularization actually exacerbates the final ascent. We see that the final ascent is hardly visible with very small regularization, but becomes more pronounced as regularization increases. Specifically, larger regularization amplifies the advantage of intermediate widths, allowing them to achieve lower test loss. In Section 4.5, we show a similar observation for robust learning algorithms, which can be viewed as using very strong regularization.

Figure 3: With stronger regularization, the optimal width increases and achieves lower loss, making the final ascent more pronounced.

**Increasing Sample Size Alleviates Final Ascent.** In our setting, although $\frac{n}{d} \to \infty$, the impact of sample size is still evident through the constant noise-to-sample-size ratio $\kappa$. Theorem 3.1 reveals that **Variance**$_{noise}$ scales with $\kappa$, rather than solely with the noise $\sigma^2$. Thus, increasing the sample size reduces the scaling of **Variance**$_{noise}$, mitigating the final ascent. We will confirm this empirically in Section 4.2.

In Section 4.1, we will conduct numerical experiments confirming that the final ascent is not limited to finite $\frac{n}{d}$. It can occur in many other settings, even when $n < d$.

## 3.2 BEYOND WIDTH: EFFECT OF DENSITY

Next, we investigate the scenario where the model width is fixed, but the model density—fraction of weights that are trainable—is decreased by masking a predetermined set of weights during training. Studying density holds importance for several reasons: Firstly, it provides the flexibility to achieve any capacity, unlike changing width which only results in (width $\times m$) parameters, where $m$ is the number of parameters at width 1. Secondly, changing density is less constrained by the model architecture, which is particularly relevant for complex architectures like InceptionResNet-v2 (which we use in our experiments in Section 4.3), where reducing width is difficult. Most importantly, as we show theoretically and empirically, density has a distinct effect on generalization compared to width. Lower density enables us to employ wider models, leading to improved generalization even under label noise.

We randomly drop a fraction of trainable parameters instead of following certain criteria as in [Jin et al., 2022], thus isolating the effect of model size from any other factors.

The setting in Theorem 3.1, where the trainable layer has a scalar output, cannot differentiate between reducing density and reducing width. To address this, we consider a three-layer linear network in which the first layer yields random features, the second layer is randomly masked and trained, and the last layer is also random. The function represented by the network is formally defined as $f(\boldsymbol{x}) = ((\boldsymbol{V} \odot \boldsymbol{M})\boldsymbol{W}\boldsymbol{x})^\top \boldsymbol{\mu}$. The first layer parameters $\boldsymbol{W}$ and input data $\boldsymbol{X}$ are the same as in Theorem 3.1. $\boldsymbol{V} \in \mathbb{R}^{q \times p}$ represents the second-layer parameters, and $\boldsymbol{M} \in \{0, 1\}^{q \times p}$ is the mask applied to $\boldsymbol{V}$. The entries in $\boldsymbol{M}$ are independently drawn from a Bernoulli distribution with parameter $\alpha \in [0, 1]$, where $\alpha$ represents the density. $\boldsymbol{\mu} \in \mathbb{R}^q$ represents the last-layer parameter, and its entries are independently drawn from $\mathcal{N}(0, 1/q)$ (the variance is set to $1/q$ so that $q$ does not appear in the final expressions). We study other variants, such as setting it to $1/d$ and letting $q = p$ in Appendix A.6. We show that the risk and its decomposition in this setting can be derived by replacing $\lambda_0$ in Theorem 3.1 with $\lambda_0/\alpha$. See the proof in Appendix A.4.

**Theorem 3.2.** *For a 3-layer linear network with $p$ hidden neurons in the first layer and random first and last layers, consider learning the second layer by ridge regression with*

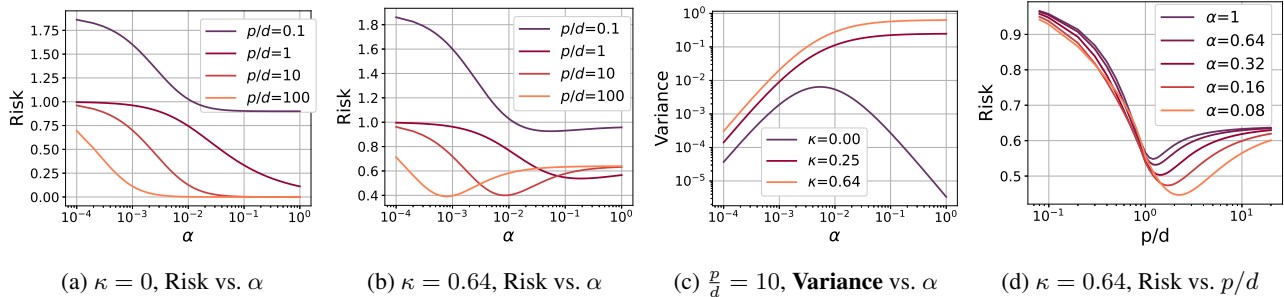

(a) $\kappa = 0$, Risk vs. $\alpha$    (b) $\kappa = 0.64$, Risk vs. $\alpha$    (c) $\frac{p}{d} = 10$, **Variance** vs. $\alpha$    (d) $\kappa = 0.64$, Risk vs. $p/d$

Figure 4: (a), (b): The risk curve changes from decreasing to U-shaped as the noise-to-sample-size ratio ($\kappa$) increases, for different values of width ($p$). (c) The total variance changes from unimodal to increasing as $\kappa$ increases. (d) Under lower density, the optimal width tends to be larger, and achieves lower test loss compared to the optimal width at higher density.

*random masks drawn from $Bernoulli(\alpha)$. Let $\lambda$, $n$ and $\sigma$ be the ridge regression parameter, number of training examples and noise level, respectively. Let $\lambda = \frac{n}{d}\lambda_0$ and $\sigma^2 = \frac{n}{d}\kappa$ for some fixed $\lambda_0$ and $\kappa$. The asymptotic expressions (where $n, d, p \to \infty$ with $\frac{n}{d} = \infty$ and $\frac{p}{d} = \gamma$) of Risk, Bias$^2$, Variance$_{clean}$ and Variance$_{noise}$ are given by their counterparts in the setting of Theorem 3.1 with $\lambda_0$ substituted with $\lambda_0/\alpha$.*

**Reducing Density Can Improve Generalization Under Label Noise.** In Fig 4, we set $\lambda_0$ to 0.05 and plot the risk and variance based on Theorem 3.2. When there is no noise, we observe a decrease in risk as density ($\alpha$) increases (Fig 4a), accompanied by a unimodal behavior of the variance (Fig 4c). However, when the noise-to-sample size ratio ($\kappa$) is sufficiently large, the risk exhibits a U-shape (Fig 4b), while the variance monotonically increases (Fig 4d).

### 3.3 ADVANTAGE OF WIDER BUT SPARSER MODELS

Theorem 3.2 demonstrates that reducing density is equivalent to a stronger $\ell_2$ regularization. Although in Section 4.3 we will empirically demonstrate that reducing density has effects beyond $\ell_2$ regularization for neural networks, it is already evident here that reducing density has a different effect than reducing width, even though both control the number of parameters. Furthermore, we observe from Figure 4b that the optimal density at a fixed width results in lower test loss as the width increases. Additionally, Figure 4d shows that the optimal width for a given density increases as the density decreases, yielding lower test loss. This highlights the advantages of wider but sparser networks.

## 4 EXPERIMENTS

### 4.1 FINAL ASCENT IN RANDOM FEATURE RIDGE REGRESSION WITH DIFFERENT $\frac{n}{d}$'S

In section 3.2, we theoretically showed that the final ascent can occur in the limit where $\frac{n}{d} \to \infty$, while $\frac{p}{d}$ remains con-

Table 1: Shape and scale of $\frac{\text{Variance}_{\text{noise}}}{\sigma^2}$ for different $n/d$ and $\sigma$. Scale is the value of $\frac{\text{Variance}_{\text{noise}}}{\sigma^2}$ at $p = 10^4$.

|  |  | $n/d$ | 0.2 | 0.5 | 1 | 2 | 4 |
|---|---|---|---|---|---|---|---|
| $\lambda$=0.01 | Shape | | ↗↘ | ↗↘ | ↗ | ↗ | ↗ |
| | Scale | | 0.26 | 1.0 | 50.2 | 1.0 | 0.33 |
| $\lambda$=0.2 | Shape | | ↗↘ | ↗ | ↗ | ↗ | ↗ |
| | Scale | | 0.25 | 1.0 | 10.5 | 1.0 | 0.33 |

stant. However, a natural question arises regarding whether the same conclusion holds when $\frac{n}{d}$ is finite. Characterizing the variance exactly is extremely troublesome when $\lambda > 0$, as it involves finding solutions for a complicated fourth-degree equation when and taking derivatives, as shown by [Adlam and Pennington, 2020a]. Hence, we conduct numerical experiments with finite values of $d$ and $n$ with varying $\frac{n}{d}$. Our results indicate that the final ascent can occur in many other settings, even when $n < d$. We plot the test loss (Figure 5) and total variance (Figure 17) against $p$. Legends show the values of $\sigma^2$, and titles show the values of $n/d$. We can clearly observe the final ascent when $\frac{n}{d} = 0.5, 1, 2, 4$.

**Effects of $n/d$ and $\lambda$ on the shape of Variance$_{\text{noise}}$.** Table 1 summarizes the shape and scale of $\frac{1}{\sigma^2}$**Variance**$_{\text{noise}}$ (i.e., $\frac{1}{d}\mathbb{E}_{\boldsymbol{X}, \boldsymbol{W}}\|\boldsymbol{A}\|_F^2$ according to Eq. 2) w.r.t. $p$ for different values of $\frac{n}{d}$ and $\lambda$ (plots are in Figures 18 and 19). Note that according to the decomposition in Eq. 2, whether the final ascent can possibly occur depends solely on if $\frac{1}{\sigma^2}$**Variance**$_{\text{noise}}$ increases in the end. We make the following observations: (1) Neither $n/d$ nor $\lambda$ alone can determine the shape of $\frac{1}{\sigma^2}$**Variance**$_{\text{noise}}$; (2) When both $\frac{n}{d}$ and $\lambda$ are small, $\frac{1}{\sigma^2}$**Variance**$_{\text{noise}}$ is unimodal, and the final ascent never occurs; (3) The effect of $n$ on the occurrence of the final ascent is non-monotonic: when $\frac{n}{d} \leq 1$, increasing $n$ turns $\frac{1}{\sigma^2}$**Variance**$_{\text{noise}}$ from unimodal to monotonically increasing, leading to the final ascent. However, when $n/d > 1$, increasing $n$ scales down $\frac{1}{\sigma^2}$**Variance**$_{\text{noise}}$ while preserving its monotonically increasing shape, resulting in the final ascent being less pronounced. This second half of the observation is captured by Theorem 3.1 showing that **Variance**$_{\text{noise}}$ scales with $\frac{\sigma^2}{n/d}$. These observations reveal a

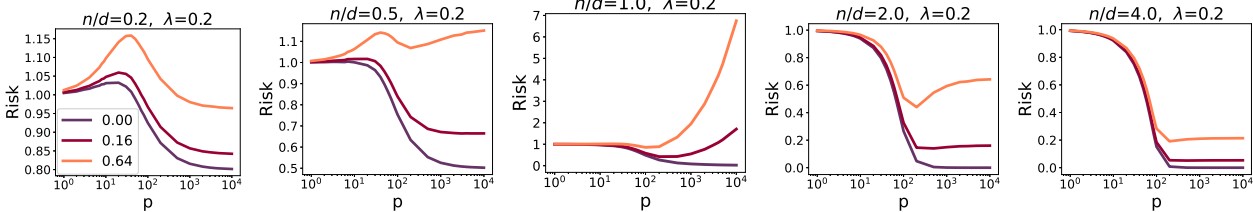

Figure 5: Final ascent in random feature ridge regression with Different $\frac{n}{d}$ Ratios. We plot the test loss while fixing $d = 100$ and $\lambda = 0.2$, and varying $\sigma^2$. Legends show the values of $\sigma^2$, and titles show the values of $n/d$.

complex behavior of the variance and highlight the need for future theoretical research to fully explain it.

### 4.2 FINAL ASCENT IN NNS: EFFECT OF WIDTH

Next, we empirically demonstrate the occurrence of the final ascent in neural networks trained with label noise across various settings. Although our theoretical results (Section 3.1) are based on regression, our empirical findings confirm that the theoretical insights hold for classification with neural networks.

We conduct a comprehensive set of experiments to thoroughly investigate the factors influencing the final ascent. Our study involve 3 datasets: MNIST [LeCun, 1998], CIFAR-10, and CIFAR-100 [Krizhevsky et al., 2009]. We train two-layer ReLu networks on MNIST, and train ResNet34 [He et al., 2016] and ViT [Dosovitskiy et al., 2020] on CIFAR-10/100. We examine two types of loss functions: mean squared error (MSE) loss and cross-entropy (CE) loss. MSE loss enables more accurate measurement of the bias and variance in test loss [Yang et al., 2020] (see Appendix C.2 ), allowing us to empirically observe the transition in the shape of variance shown in Thm 3.1. We consider two types of noise: symmetric noise generated through random label flipping and asymmetric noise generated in a class-dependent manner (see Appendix C.1). Asymmetric noise better resembles real-world noise distributions [Patrini et al., 2017]. Further experimental details can be found in Appendix C.

**Final Ascent Occurs in Various Settings.** Fig 6 to 9 demonstrate the presence of final ascent across architectures, loss functions, and noise types. E.g., in Fig 6 and 7, as the noise level increases, we observe a transition in the loss curve from double descent or monotonically decreasing to a curve with final ascent. The corresponding test accuracy plots can be found in Appendix D.2. Results for ViT are in Fig 20.

**Transition of the Variance Shape as Label Noise Increases.** As discussed in Sec 3.1, the final ascent in the test loss is attributed to the increasing-decreasing-increasing pattern of the variance. The results in Fig 6c and 7c confirm that this holds true for NNs. The shape of the variance curve turns from unimodal to increasing-decreasing-increasing as

the noise increases, matching our theoretical results.

**Stronger $\ell_2$ Regularization Exacerbates Final Ascent.** Our theoretical analysis in Sec 3.1 demonstrates that stronger regularization exacerbates final ascent. This finding is further corroborated by our experimental results, as shown by comparing Fig 6a and 6b, as well as Fig 7a and 7b. For example, on CIFAR-10 with 20% noise, the test loss exhibits the well-known double descent when $\lambda = 0.0005$ (the parameter of $l_2$), while it demonstrates the final ascent phenomenon when $\lambda = 0.001$. Note that using regularization often yields a better generalization.

**Asymmetric Noise May Exacerbate Final Ascent.** Fig 8 shows that asymmetric noise significantly exacerbates the final ascent on CIFAR-10. Under 40% noise, the final ascent is absent for symmetric noise but present for asymmetric noise. However, this trend is not obvious on CIFAR-100, possibly due to the different noise generation process (Appendix C.1) where the noise on CIFAR-100 is less skewed.

**Larger Sample Size Alleviates Final Ascent.** As shown in Thm 3.1, the scale of **Variance**$_{\text{noise}}$ is determined by the noise-to-sample-size ratio rather than the noise level itself, implying that increasing the sample size can counteract the impact of noise and alleviate final ascent. Our experiments further confirm this. Fig 9 shows the test loss with varying sample sizes on MNIST (MSE loss) and CIFAR-10 (CE loss). The final ascent occurs when only $1/20$ of the original data are used, but not when $1/2$ are used on MNIST and $1/5$ on CIFAR-10. This is also observed with ViTs (Fig 20).

**Required Noise Level for Final Ascent.** Based on the preceding discussions, it is evident that the required noise level depends on several factors, including the degree of regularization, the sample size, and the nature of the noise distribution. In general, a lower noise level is required with strong regularization, smaller sample sizes, and strong noise types. **Notably, the required noise level can be as low as 20% (Fig 6, 7). On CIFAR-10, even with full data and the commonly used small regularization ($5e^{-4}$), final ascent occurs under only 30% asymmetric noise (Fig 8).** In practice, where the noise typically resembles asymptotic noise and strong regularization is applied in the presence of noise, final ascent is highly likely to occur under low noise levels.

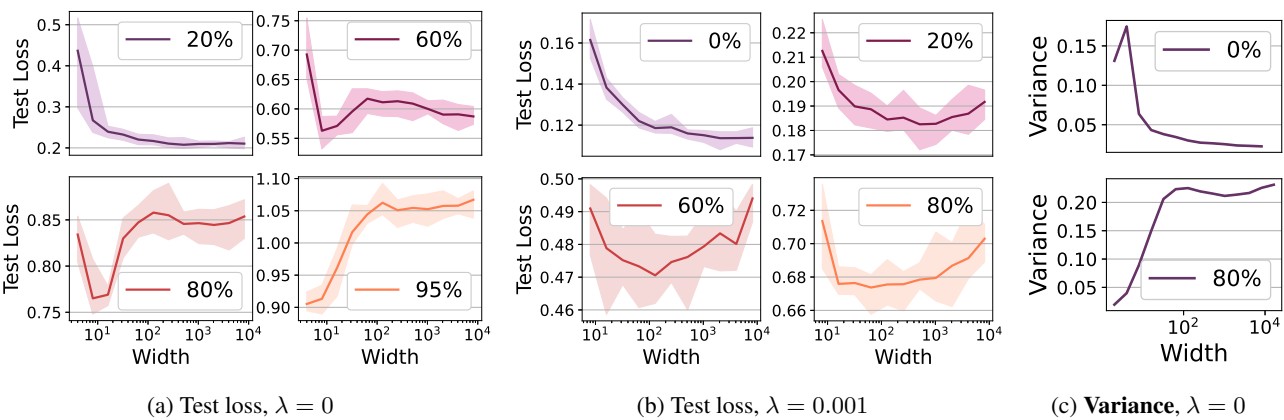

Figure 6: Test loss and total variance on MNIST using MSE loss with (a) $\lambda = 0$ and (b) $\lambda = 0.001$ ($l_2$ regularization). We split the original dataset into 20 subsets to measure bias and variance (see details Appendix C.2). Final ascent occurs when the noise level reaches a certain threshold. Stronger regularization exacerbates the final ascent phenomenon and reduces the required noise level. (c) As the noise level increases, the variance transitions from a unimodal shape to an increasing-decreasing-increasing pattern.

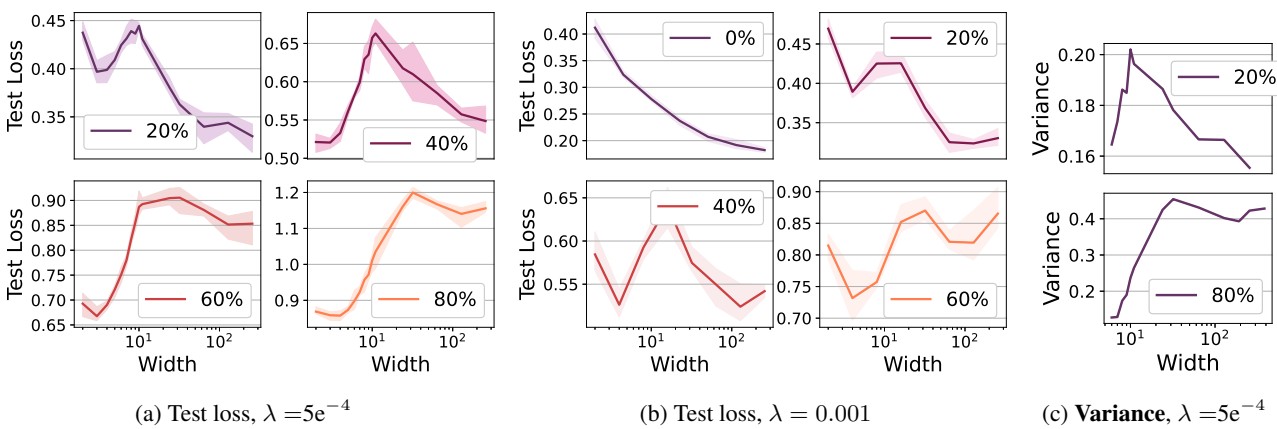

Figure 7: Test loss and total variance on CIFAR-10 using MSE loss with (a) $\lambda = 0.0005$ and (b) $\lambda = 0.001$. We split the original dataset into 5 subsets to measure bias and variance (see details Appendix C.2). The observed patterns are consistent with Figure 6, confirming that stronger regularization exacerbates the final ascent and that increasing noise causes the transition in the variance shape.

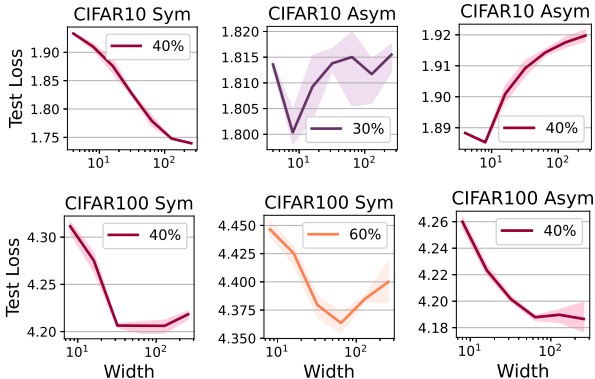

Figure 8: Results under symmetric and asymmetric noise on full CIFAR-10/100 using CE loss. $\lambda = 0.0005$.

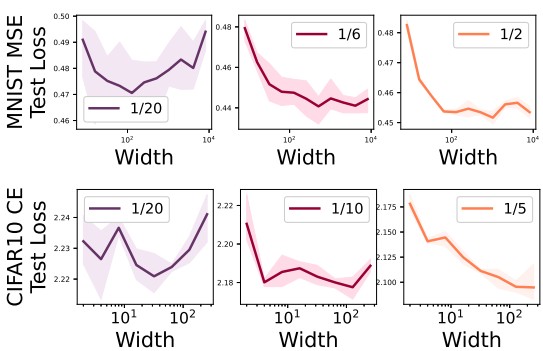

Figure 9: Test loss obtained with different sample sizes for MNIST using MSE loss (top) and CIFAR-10 using CE loss (bottom). We use $\lambda=0.001$ and 60% noise for both. Legends indicate the fraction of data used for training.

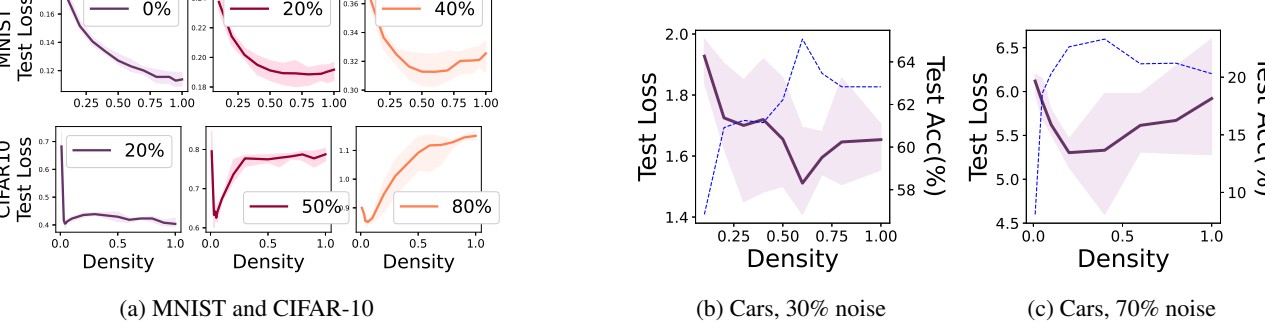

(a) MNIST and CIFAR-10

(b) Cars, 30% noise

(c) Cars, 70% noise

Figure 10: Test performance models with varied densities on (a) MNIST and CIFAR-10 and (b)(c) Red Stanford Car. In (a) legends indicate noise ratios. We observe that the test loss curve changes from a decreasing trend to a U-shape as noise increases. In (b)(c) the purple solid line shows test loss and the blue dashed line shows accuracy.

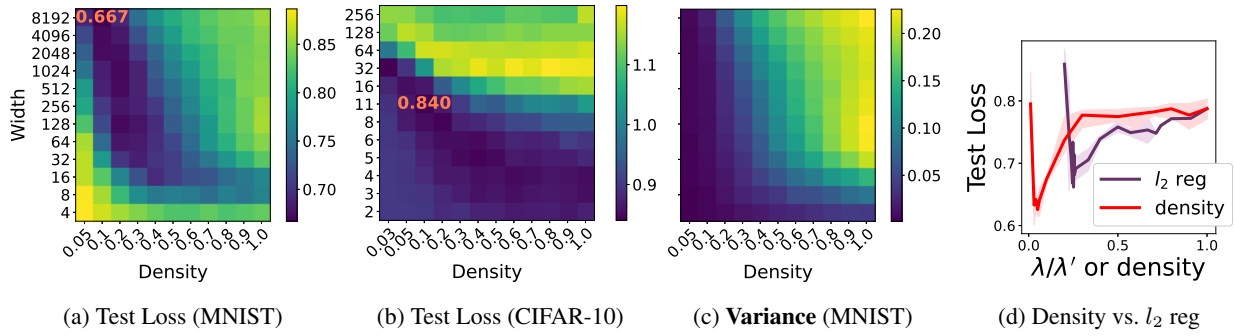

(a) Test Loss (MNIST)

(b) Test Loss (CIFAR-10)

(c) **Variance** (MNIST)

(d) Density vs. $l_2$ reg

Figure 11: Test loss (a)(b) and variance (c) of models at varied widths and densities. (d) compares the effects density and $l_2$ regularization. When varying $\lambda$, density is fixed to 1. When varying density, $\lambda$ is fixed to $\lambda' = 0.0005$. X-axis represents the inverse of the scale-up of $\lambda$, i.e., $\lambda/\lambda'$, for the regularization curve (purple), and density for the density curve (red).

## 4.3 FINAL ASCENT IN NNS: EFFECT OF DENSITY

Next, we study the effect of density through experiments on three datasets: MNIST, CIFAR-10, and Red Stanford Cars [Jiang et al., 2020] containing real-world label noise (see Appendix C.1). We train an InceptionResNet-v2 on Red Stanford Cars . Since the InceptionResNet-v2 architecture is intricate, there is no straightforward way to vary its width, and demands reducing density. Other details are in Appendix C.

**Reducing Density Improves Generalization.** Fig 10 shows that when the label noise is small (e.g., 0% on MNIST, 20% on CIFAR-10), test loss improves as density increases. In contrast, when the label noise is sufficiently large (e.g., 40% on CIFAR-10, 50% on MNIST, 30%/70% on Red Stanford Car), lowest test loss is achieved at an intermediate density.

**Reducing Density has an Effect Beyond $\ell_2$ Regularization for NNs.** Thm 3.2 shows that reducing density is equivalent to increasing the $\ell_2$ regularization by the inverse factor for random feature ridge regression. However, Fig 11d (test accuracy is in Appendix D.3) shows that reducing density of neural networks has an effect beyond $\ell_2$ regularization. Although both loss curves exhibit a U shape, bottom of the

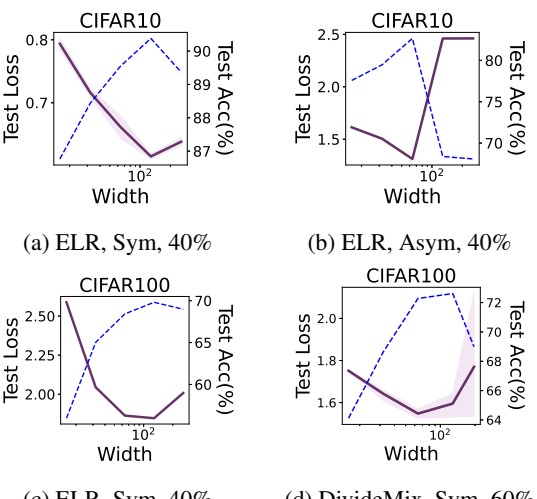

(a) ELR, Sym, 40%

(b) ELR, Asym, 40%

(c) ELR, Sym, 40%

(d) DivideMix, Sym, 60%

Figure 12: Test loss (solid) and accuracy (dashed) of models trained with ELR and DivideMix on CIFAR-10/100. Comparing (a) with Fig 9, we see that ELR even exacerbates the final ascent.

U shape for density is lower than that of varying $\ell_2$ regularization. In other words, adjusting density can achieve a lower loss than adjusting $\ell_2$ regularization.

## 4.4 ADVANTAGE OF WIDER BUT SPARSER MODELS

Fig 11c shows an increasing trend in variance towards the top-right corner. Fig 11a and 11b show that models with low loss (darkest blue) are distributed along the diagonal and the lowest loss (annotated in red) is achieved with models with very small density. This confirms our theoretical results in Section 3.2 that models of optimal width under smaller density exhibit better generalization, highlighting the advantage of employing wider yet sparser models to address label noise. Test accuracy and bias are shown in Appendix D.3.

## 4.5 FINAL ASCENT IN NEURAL NETWORKS: ROBUST ALGORITHMS

We study the effect of robust algorithms, which are typically employed in the presence of label noise. We consider two SOTA algorithms, ELR [Liu et al., 2020] and DivideMix [Li et al., 2020] on CIFAR-10/100 (details in Appendix C.4). Interestingly, we see that the final ascent can still be observed (Fig 12 and 29), and in some cases is even exacerbated. For example, without ELR, the final ascent is not observed on CIFAR-10 under 60% symmetric noise with $1/5$ of the data (Fig 9). In contrast, when ELR is applied, the final ascent occurs under only 40% symmetric noise on the full data. Additional experiments are in Appendix D.4. The final ascent regarding model density is also observed in Fig 30 to 32.

**Connection to $\ell_2$ Regularization.** The presence of the final ascent under robust algorithms aligns with our theoretical and empirical findings that stronger $\ell_2$ regularization exacerbates this phenomenon. As robust algorithms act as a form of stronger regularization compared to $\ell_2$, it is logical that they amplify the final ascent rather than mitigating it.

**Other Robust Algorithms against Noisy Labels.** In general, all robust methods can be thought of as a form of implicit regularization at a very high level, since they share the common intuition of preventing the model from fitting certain data too closely. For example, [Zhang and Sabuncu, 2018, Jiang et al., 2018, Xia et al., 2019, Mirzasoleiman et al., 2020] either explicitly or implicitly put more weight on clean data, making the model fit the noisy data less. Therefore, we believe the same conclusions would likely apply to these other methods as well.

Another robust algorithm, particularly related to density, is [Xia et al., 2020], which divides the model weights into two sets, updating one while shrinking the other. The division is dynamically adjusted during training to combat label noise. This implies that learning fewer weights can benefit robustness. Our findings make an even stronger statement: robustness can be enhanced by simply removing some weights randomly from the very beginning of training, provided the weight removal ratio is appropriate. The implication of our

research is that naively dropping weights is a baseline worth considering in this context. Additionally, we provide theoretical groundwork for understanding the effects of methods similar to [Xia et al., 2020], showing that much of the benefit can be viewed as simply reducing the hypothesis space. From the bias-variance perspective, reducing density can lead to a smaller noise-dependent variance.

## 5 CONCLUSION AND DISCUSSION

We present the *final ascent*, an intriguing phenomenon, w.r.t. both model width and density. Our comprehensive theoretical and empirical investigations yield crucial insights, including the transition in variance shape, the interplay between width and density, and the roles of sample size, regularization, and robust methods. Our results highlight (1) the need for caution when using large models, (2) lower density combined with wider models improves generalization even under noise, and (3) the amplified significance of these considerations in scenarios with limited data, or under strong regularization/robust methods.

## ACKNOWLEDGMENTS

This research was partially supported by the National Science Foundation CAREER Award 2146492.

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

## A THEORETICAL RESULTS

**Notations:** We use bold-faced letter for matrix and vectors. The training dataset is denoted by $(\boldsymbol{X}, \boldsymbol{y})$ where $\boldsymbol{X} = [\boldsymbol{x}_1, \boldsymbol{x}_2, \ldots, \boldsymbol{x}_n]$ and $\boldsymbol{y} = [y_1, y_2, \ldots, y_n]^\top$. Each input vector $\boldsymbol{x}_i \in \mathbb{R}^d$ is independently drawn from a Gaussian distribution $\mathcal{N}(0, \mathbf{I}_d/d)$. Each label $y_i \in \mathbb{R}$ is generated by $y_i = \boldsymbol{x}_i^\top \boldsymbol{\theta} + \epsilon_i$. We assume $\boldsymbol{\theta} \in \mathbb{R}^d$ has its entries independently drawn from $N(0, 1)$. $\epsilon_i \in \mathbb{R}$ is the label noise drawn from $N(0, \sigma^2)$ for each $\boldsymbol{x}_i$. We assume each test example $(\boldsymbol{x}, y)$ is clean, i.e., $y = \boldsymbol{x}^\top \boldsymbol{\theta}$.

### A.1 BIAS-VARIANCE DECOMPOSITION OF THE MSE LOSS IN SECTION 3.1

Here we show the derivation of Equation 1.

$$
\begin{aligned}
\mathbf{Bias}^2 &= \mathbb{E}_{\boldsymbol{\theta}} \mathbb{E}_{\boldsymbol{x}} (\mathbb{E}_{\boldsymbol{X}, \boldsymbol{W}, \epsilon} f(\boldsymbol{x}) - y)^2 \\
&= \mathbb{E}_{\boldsymbol{\theta}} \mathbb{E}_{\boldsymbol{x}} (\mathbb{E}_{\boldsymbol{X}, \boldsymbol{W}, \epsilon} ((\boldsymbol{W}\boldsymbol{x})^\top \hat{\boldsymbol{\beta}}) - y)^2 \\
&= \mathbb{E}_{\boldsymbol{\theta}} \mathbb{E}_{\boldsymbol{x}} [\boldsymbol{x}^\top (\mathbb{E}_{\boldsymbol{X}, \boldsymbol{W}, \epsilon} (\boldsymbol{B}\boldsymbol{\theta} + \boldsymbol{A}\epsilon) - \boldsymbol{\theta})]^2 \\
&= \mathbb{E}_{\boldsymbol{\theta}} \mathbb{E}_{\boldsymbol{x}} \mathrm{Tr}[(\mathbb{E}_{\boldsymbol{X}, \boldsymbol{W}, \epsilon} (\boldsymbol{B}\boldsymbol{\theta}) - \boldsymbol{\theta})^\top \boldsymbol{x}\boldsymbol{x}^\top (\mathbb{E}_{\boldsymbol{X}, \boldsymbol{W}, \epsilon} (\boldsymbol{B}\boldsymbol{\theta}) - \boldsymbol{\theta})] \\
&= \mathbb{E}_{\boldsymbol{\theta}} \mathrm{Tr}[(\mathbb{E}_{\boldsymbol{X}, \boldsymbol{W}, \epsilon} (\boldsymbol{B}\boldsymbol{\theta}) - \boldsymbol{\theta})^\top \mathbb{E}_{\boldsymbol{x}} (\boldsymbol{x}\boldsymbol{x}^\top) (\mathbb{E}_{\boldsymbol{X}, \boldsymbol{W}, \epsilon} (\boldsymbol{B}\boldsymbol{\theta}) - \boldsymbol{\theta})] \\
&= \frac{1}{d} \mathbb{E}_{\boldsymbol{\theta}} \|\mathbb{E}_{\boldsymbol{X}, \boldsymbol{W}, \epsilon} (\boldsymbol{B}\boldsymbol{\theta}) - \boldsymbol{\theta}\|_F^2 \\
&= \frac{1}{d} \mathbb{E}_{\boldsymbol{\theta}} \mathrm{Tr}[(\mathbb{E}_{\boldsymbol{X}, \boldsymbol{W}} \boldsymbol{B} - \mathbf{I}) \boldsymbol{\theta} \boldsymbol{\theta}^\top (\mathbb{E}_{\boldsymbol{X}, \boldsymbol{W}} \boldsymbol{B} - \mathbf{I})] \\
&= \frac{1}{d} \mathrm{Tr}[(\mathbb{E}_{\boldsymbol{X}, \boldsymbol{W}} \boldsymbol{B} - \mathbf{I}) \mathbb{E}_{\boldsymbol{\theta}} (\boldsymbol{\theta} \boldsymbol{\theta}^\top) (\mathbb{E}_{\boldsymbol{X}, \boldsymbol{W}} \boldsymbol{B} - \mathbf{I})] \\
&= \frac{1}{d} \|\mathbb{E}_{\boldsymbol{X}, \boldsymbol{W}} \boldsymbol{B} - \mathbf{I}\|_F^2
\end{aligned}
$$

$$
\begin{aligned}
\mathbf{Variance} &= \mathbb{E}_{\boldsymbol{\theta}} \mathbb{E}_{\boldsymbol{x}} \mathbb{V}_{\boldsymbol{X}, \boldsymbol{W}, \epsilon} f(\boldsymbol{x}) \\
&= \mathbb{E}_{\boldsymbol{\theta}} \mathbb{E}_{\boldsymbol{x}} \mathbb{E}_{\boldsymbol{X}, \boldsymbol{W}, \epsilon} [\boldsymbol{x}^\top (\boldsymbol{B}\boldsymbol{\theta} + \boldsymbol{A}\epsilon) - \mathbb{E}_{\boldsymbol{X}, \boldsymbol{W}, \epsilon} \boldsymbol{x}^\top (\boldsymbol{B}\boldsymbol{\theta} + \boldsymbol{A}\epsilon)]^2 \\
&= \frac{1}{d} \mathbb{E}_{\boldsymbol{\theta}} \mathbb{E}_{\boldsymbol{X}, \boldsymbol{W}, \epsilon} \|(\boldsymbol{B}\boldsymbol{\theta} + \boldsymbol{A}\epsilon) - \mathbb{E}_{\boldsymbol{X}, \boldsymbol{W}} \boldsymbol{B}\boldsymbol{\theta}\|_F^2 \\
&= \frac{1}{d} \mathbb{E}_{\boldsymbol{\theta}} \mathbb{E}_{\boldsymbol{X}, \boldsymbol{W}, \epsilon} \|(\boldsymbol{B} - \mathbb{E}_{\boldsymbol{X}, \boldsymbol{W}} \boldsymbol{B}) \boldsymbol{\theta} + \boldsymbol{A}\epsilon\|_F^2 \\
&= \frac{1}{d} \mathbb{E}_{\boldsymbol{\theta}} \mathbb{E}_{\boldsymbol{X}, \boldsymbol{W}} \|(\boldsymbol{B} - \mathbb{E}_{\boldsymbol{X}, \boldsymbol{W}} \boldsymbol{B}) \boldsymbol{\theta}\|_F^2 + \frac{1}{d} \mathbb{E}_{\boldsymbol{X}, \boldsymbol{W}, \epsilon} \|\boldsymbol{A}\epsilon\|_F^2 \\
&= \underbrace{\frac{1}{d} \mathbb{E}_{\boldsymbol{X}, \boldsymbol{W}} \|\boldsymbol{B} - \mathbb{E}_{\boldsymbol{X}, \boldsymbol{W}} \boldsymbol{B}\|_F^2}_{\mathbf{Variance}_{\text{clean}}} + \underbrace{\frac{\sigma^2}{d} \mathbb{E}_{\boldsymbol{X}, \boldsymbol{W}} \|\boldsymbol{A}\|_F^2}_{\mathbf{Variance}_{\text{noise}}}
\end{aligned}
$$

### A.2 EXPRESSIONS OF BIAS AND Variance_clean

Yang et al. [2020] analyzed the bias-variance decomposition without considering label noise. Hence the variance in their analysis corresponds to **Variance**$_{\text{clean}}$ in ours. We show their expressions of **Bias**$^2$ and **Variance**$_{\text{clean}}$ below, based on which we plot the dashed gray line in Figure 1 and the solid gray line in Figure 2.

$$
\mathbf{Bias}^2 = \frac{1}{4} \Phi_3(\lambda_0, \gamma)^2
$$

$$
\mathbf{Variance}_{\text{clean}} = \begin{cases} \frac{\Phi_1(\lambda_0, \gamma)}{2\Phi_2(\lambda_0, \gamma)} - \frac{(1-\gamma)(1-2\gamma)}{2\gamma} - \frac{1}{4}\Phi_3(\lambda_0, \gamma)^2, & \gamma \le 1, \\ \frac{\Phi_1(\lambda_0, 1/\gamma)}{2\Phi_2(\lambda_0, 1/\gamma)} - \frac{\gamma-1}{2} - \frac{1}{4}\Phi_3(\lambda_0, \gamma)^2, & \gamma > 1, \end{cases}
$$

where

$$
\begin{aligned}
\Phi_1(\lambda_0, \gamma) &= \lambda_0(\gamma + 1) + (\gamma - 1)^2, \\
\Phi_2(\lambda_0, \gamma) &= \sqrt{(\lambda_0 + 1)^2 + 2(\lambda_0 - 1)\gamma + \gamma^2}, \\
\Phi_3(\lambda_0, \gamma) &= \Phi_2(\lambda_0, \gamma) - \lambda_0 - \gamma + 1.
\end{aligned}
$$

## A.3   PROOF OF THEOREM 3.1

**Lemma A.1.** *Define* $\tilde{B} = W^\top(WW^\top + \lambda_0 I)^{-1} W$. $\|\tilde{B}\tilde{B}^\top - \frac{n}{d}AA^\top\|_2 = 0$ *almost surely, i.e.,* $\mathbb{P}(\|\tilde{B}\tilde{B}^\top - \frac{n}{d}AA^\top\|_2 \leq \epsilon) \geq 1 - \delta$ *where $\epsilon$ and $\delta$ both tend to 0 under the asymptotics $n \to \infty$, $d \to \infty$ and $n/d \to \infty$.*

*Proof.*  Define

$$\Delta := \frac{d}{n}XX^\top - I,$$
$$\Psi := (WW^\top + \lambda_0 I)^{-1}$$
$$\Gamma := (\frac{d}{n}WXX^\top W^\top + \lambda_0 I)^{-1}$$
$$\Omega := \Gamma - \Psi.$$

Then we have

$$\tilde{B}\tilde{B}^\top - \frac{n}{d}AA^\top = \Omega W \Delta W^\top \Omega + \Psi W \Delta W^\top \Psi + \Omega W \Delta W^\top \Psi + \Psi W \Delta W^\top \Omega$$
$$+ \ \Omega W W^\top \Omega + \Omega W W^\top \Psi + \Psi W W^\top \Omega.$$

By triangle inequality and the sub-multiplicative property of spectral norm we have

$$\|\tilde{B}\tilde{B}^\top - \frac{n}{d}AA^\top\|_2 \leq \|W\|_2^2\|\Omega\|_2^2\|\Delta\|_2 + \|W\|_2^2\|\Psi\|_2^2\|\Delta\|_2 + 2\|W\|_2^2\|\Psi\|_2\|\Omega\|_2\|\Delta\|_2$$
$$+ \ \|W\|_2^2\|\Omega\|_2^2 + 2\|W\|_2^2\|\Psi\|_2\|\Omega\|_2$$

It remains to show that with the asymptotic assumption $n/d \to \infty$, $\|\Omega\|_2 = 0$ and $\|\Delta\|_2 = 0$ almost surely, and $\|W\|_2$ and $\|\Psi\|$ can be bounded from above Yang et al. [2020]:

$$\mathbb{P}(\|\Delta\|_2 \leq 4\sqrt{\frac{d}{n}} + 4\frac{d}{n}) \geq 1 - e^{-d/2}$$
$$\|\Omega\|_2 \leq \|\Psi\|_2^2\|W\|_2^2\|\Delta\|_2 + O(\|\Delta\|_2)$$
$$\|W\|_2 \stackrel{a.s.}{=} 1 + \sqrt{\eta} < \infty$$
$$\|\Psi\|_2 \leq \frac{1}{\lambda_0} < \infty.$$

Therefore we have $\|\tilde{B}\tilde{B}^\top - \frac{n}{d}AA^\top\|_2 = 0$ almost surely.                    □

**Corollary A.2.**  $\frac{\sigma^2}{d}\|A\|_F^2 = \frac{\kappa}{d}\|\tilde{B}\|_F^2$ *almost surely.*

*Proof.*  By lemma A.1 we have

$$|\operatorname{Tr}(\frac{1}{n}\tilde{B}\tilde{B}^\top - \frac{1}{d}AA^\top)| = \frac{1}{n}\operatorname{Tr}(\tilde{B}\tilde{B}^\top - \frac{n}{d}AA^\top)$$
$$\leq \frac{d}{n}\|\tilde{B}\tilde{B}^\top - \frac{n}{d}AA^\top\|_2$$
$$= 0,$$

which yields $\frac{1}{d}\|A\|_F^2 = \frac{1}{n}\|\tilde{B}\|_F^2$ and thus $\frac{\sigma^2}{d}\|A\|_F^2 = \frac{\kappa}{d}\|\tilde{B}\|_F^2$.                    □

Now it only remains to compute $\frac{1}{d}\|\tilde{B}\|_F^2$. By Sherman–Morrison formula,

$$\tilde{B} = I - (I + \frac{\alpha}{\eta}Q)^{-1},$$

where $\alpha = \lambda_0^{-1}$, $\boldsymbol{Q} = (d/p)\boldsymbol{W}^\top \boldsymbol{W}$ and $\eta = d/p = 1/\gamma$. Let $F^{\boldsymbol{Q}}$ be the empirical spectral distribution of $\boldsymbol{Q}$, i.e.,

$$F^{\boldsymbol{Q}}(x) = \frac{1}{d}\#\{j \le d : \lambda_j \le x\},$$

where $\#S$ denotes the cardinality of the set $S$ and $\lambda_j$ denotes the $j$-th eigenvalue of $\boldsymbol{Q}$. Then

$$\frac{\|\tilde{\boldsymbol{B}}\|^2}{d} = \int_{\mathbb{R}^+} \frac{(\frac{\alpha}{\eta}x)^2}{(1+\frac{\alpha}{\eta}x)^2} dF^{\boldsymbol{Q}}(x).$$

By Marchenko-Pastur Law Bai and Silverstein [2010],

$$\begin{aligned}
\frac{\|\tilde{\boldsymbol{B}}\|^2}{d} &= \frac{1}{2\pi}\int_{\eta_-}^{\eta_+} \frac{\sqrt{(\eta_+ - x)(x - \eta_-)}(\frac{\alpha}{\eta}x)^2}{\eta x(1+\frac{\alpha}{\eta}x)^2} dx \\
&= \frac{1}{2\alpha}\left(\frac{-\alpha^2/\eta^2 - (3\alpha - 2\alpha^2)/\eta - \alpha^2 - 3\alpha - 2}{\sqrt{\alpha^2/\eta^2 + (2\alpha - 2\alpha^2)/\eta + \alpha^2 + 2\alpha + 1}} + \alpha/\eta + \alpha + 2\right).
\end{aligned} \tag{3}$$

Combining equation 3 and corollary A.2 and substituting $\alpha = \lambda_0^{-1}, \eta = \gamma^{-1}$ into the result completes the proof.

## A.4 PROOF OF THEOREM 3.2

Define the following shorthand

$$\begin{aligned}
\boldsymbol{F} &:= \boldsymbol{W}\boldsymbol{X} \\
\boldsymbol{D}_i &:= \mathbf{diag}(\boldsymbol{M}_{i,1}, \boldsymbol{M}_{i,2}, \ldots, \boldsymbol{M}_{i,p}) \\
\mu_i &:= i\text{-th entry of } \boldsymbol{\mu} \\
\boldsymbol{V}_i^\top &:= i\text{-th row of } \boldsymbol{V} \\
\boldsymbol{\Sigma} &:= \sum_{i=1}^q \mu_i^2 \boldsymbol{D}_i \\
\boldsymbol{H} &:= \boldsymbol{y}^\top \boldsymbol{F}^\top \boldsymbol{\Sigma} \boldsymbol{F}(\boldsymbol{F}^\top \boldsymbol{\Sigma} \boldsymbol{F} + \lambda \mathbf{I})^{-1}
\end{aligned}$$

The parameter of the second layer is given by ridge regression:

$$\begin{aligned}
\hat{\boldsymbol{V}} &= \arg\min_{\boldsymbol{V}\in\mathbb{R}^{q\times p}} \|\left((\boldsymbol{V}\odot\boldsymbol{M})\boldsymbol{W}\boldsymbol{X}\right)^\top \boldsymbol{\mu} - \boldsymbol{y}\|_F^2 + \lambda\|\boldsymbol{V}\|_F^2 \\
&= \arg\min_{\boldsymbol{V}\in\mathbb{R}^{q\times p}} \|\sum_{i=1}^q \mu_i \boldsymbol{V}_i^\top \boldsymbol{D}_i \boldsymbol{F} - \boldsymbol{y}^\top\|_F^2 + \lambda\|\boldsymbol{V}\|_F^2.
\end{aligned}$$

Solving the above yields

$$\hat{\boldsymbol{V}}_i^\top = \frac{1}{\lambda}(\boldsymbol{y}^\top - \boldsymbol{H})\boldsymbol{F}^\top(\mu_i \boldsymbol{D}_i).$$

Given a clean test example $(\boldsymbol{x}, y)$, the expression of the risk is

$$\begin{aligned}
\mathbf{Risk} &= \mathbb{E}\|\sum_{i=1}^q \mu_i \hat{\boldsymbol{V}}_i^\top \boldsymbol{W}\boldsymbol{x} - \boldsymbol{\theta}^\top \boldsymbol{x}\|_F^2 \\
&= \mathbb{E}\|\frac{1}{\lambda}(\boldsymbol{y}^\top - \boldsymbol{H})\boldsymbol{F}^\top \boldsymbol{\Sigma}\boldsymbol{W}\boldsymbol{x} - \boldsymbol{\theta}^\top \boldsymbol{x}\|_F^2 \\
&= \mathbb{E}\|\frac{1}{\lambda}(\boldsymbol{y}^\top - \boldsymbol{y}^\top \boldsymbol{F}^\top \boldsymbol{\Sigma}\boldsymbol{F}(\boldsymbol{F}^\top \boldsymbol{\Sigma}\boldsymbol{F} + \lambda\mathbf{I})^{-1})\boldsymbol{F}^\top \boldsymbol{\Sigma}\boldsymbol{W}\boldsymbol{x} - \boldsymbol{\theta}^\top \boldsymbol{x}\|_F^2
\end{aligned}$$

Observe that the diagonal matrix $\boldsymbol{\Sigma}$, whose diagonal entries $\sum_{i=1}^q \mu_i^2 \boldsymbol{M}_{i,j} = \frac{1}{q}\sum_{i=1}^q (\sqrt{q}\mu_i)^2 \boldsymbol{M}_{i,j}$ converge in probability to $\alpha$ as $q \to \infty$, captures all the effects of $\boldsymbol{\mu}$ and $\boldsymbol{M}$ on the risk. Thus we can replace $\boldsymbol{\Sigma}$ with $\alpha\mathbf{I}_p$

$$\mathbf{Risk} = \mathbb{E}\|\frac{\alpha}{\lambda}\boldsymbol{y}^\top \boldsymbol{F}\boldsymbol{W}\boldsymbol{x} - \frac{\alpha^2}{\lambda^2}\boldsymbol{y}^\top \boldsymbol{F}^\top \boldsymbol{F}(\frac{\alpha}{\lambda}\boldsymbol{F}^\top \boldsymbol{F} + \mathbf{I})^{-1}\boldsymbol{F}^\top \boldsymbol{W}\boldsymbol{x} - \boldsymbol{\theta}^\top \boldsymbol{x}\|_F^2. \tag{4}$$

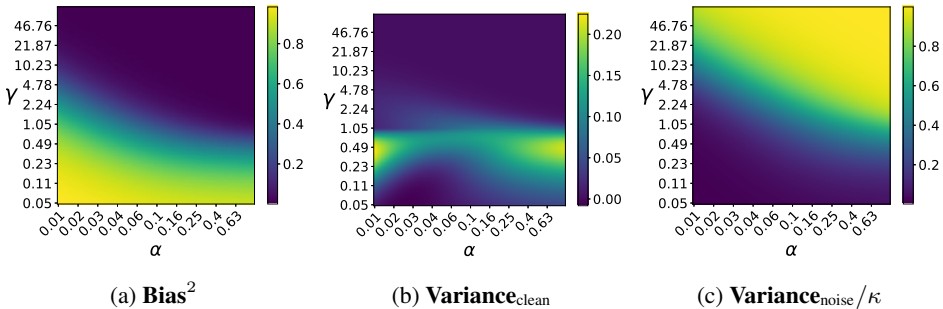

(a) **Bias**$^2$       (b) **Variance**$_{\text{clean}}$       (c) **Variance**$_{\text{noise}}/\kappa$

Figure 13: Expressions of **Bias**$^2$, **Variance**$_{\text{clean}}$ and **Variance**$_{\text{noise}}/\kappa$ in Theorem 3.2

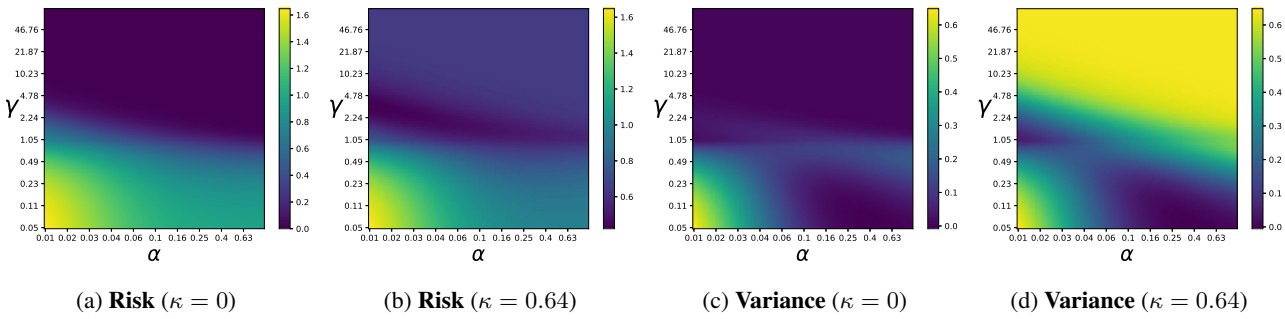

(a) **Risk** $(\kappa = 0)$    (b) **Risk** $(\kappa = 0.64)$    (c) **Variance** $(\kappa = 0)$    (d) **Variance** $(\kappa = 0.64)$

Figure 14: Expressions of the risk and variance under different noise levels with $\tau = \gamma$. We let $\lambda_0 = 0.05$.

It remains to show that the expression of the risk is exactly the same as the risk in the setting of Section 3.1 with $\lambda$ replaced by $\lambda/\alpha$. The risk in Section 3.1 (for convenience denote it by **Risk**$_0$) can be written as:

$$\begin{aligned}
\mathbf{Risk}_0 &= \mathbb{E}\|\boldsymbol{y}^\top \boldsymbol{F}^\top (\boldsymbol{F}\boldsymbol{F}^\top + \lambda \mathbf{I})^{-1}\boldsymbol{W}\boldsymbol{x} - \boldsymbol{\theta}^\top \boldsymbol{x}\|_F^2 \\
&= \mathbb{E}\|\frac{1}{\lambda}\boldsymbol{y}^\top \boldsymbol{F}\boldsymbol{W}\boldsymbol{x} - \frac{1}{\lambda^2}\boldsymbol{y}^\top \boldsymbol{F}^\top \boldsymbol{F}(\frac{1}{\lambda}\boldsymbol{F}^\top \boldsymbol{F} + \mathbf{I})^{-1}\boldsymbol{F}^\top \boldsymbol{W}\boldsymbol{x} - \boldsymbol{\theta}^\top \boldsymbol{x}\|_F^2.
\end{aligned} \tag{5}$$

Equation 5 is obtained by applying Woodbury matrix identity to $(\boldsymbol{F}\boldsymbol{F}^\top + \lambda\mathbf{I})^{-1}$. It is easy to check that replacing $\lambda$ in RHS of equation 5 with $\lambda/\alpha$ yields the same as RHS of equation 4.

## A.5   Bias$^2$ AND Variance$_{\text{noise}}/\kappa$ IN SECTION 3.2

In Figure 13 we plot the expressions of **Bias**$^2$ and **Variance**$_{\text{noise}}/\kappa$ against both width and density based on Theorem 3.2. **Bias**$^2$ monotonically decreases along both axes and **Variance**$_{\text{noise}}/\kappa$ monotonically increases along both axes. **Variance**$_{\text{clean}}$ variance is unimodal along y-axis (width), manifests more complicated behavior along x-axis (density), and decreases along both axes once width is sufficiently large. It is clear that such behavior differs from that in classical bias-variance tradeoff.

## A.6   VARIANTS OF THEOREM 3.2

In Theorem 3.2 we let $\boldsymbol{\mu}$'s entries be drawn from $\mathcal{N}(0, 1/q)$ so that $q$ does not appear in the expression of the risk. Alternatively we can let $\boldsymbol{\mu}$'s entries be drawn from $\mathcal{N}(0, 1/d)$ and assume $q/d = \tau$. Then the risk and its decomposition are dependent on $\gamma, \tau, \alpha$. Similar to the proof in A.4, we can show that in this case we only have to replace $\lambda_0$ in the setting of Theorem 3.1 with $\lambda_0/(\tau\alpha)$ to get the expressions of risk. We further let $\tau = \gamma$ and plot the risk, **Variance**, **Bias**$^2$, **Variance**$_{\text{clean}}$ (i.e., **Variance** with $\kappa = 0$), **Variance**$_{\text{noise}}$ in Figures 14 and 15.

## A.7   CONNECTION TO THE OBSERVATION MADE BY Golubeva et al. [2020]

Our Theorem 3.2 supports the empirical observation in Golubeva et al. [2020] that increasing width while fixing the number of parameters (by reducing density) improves generalization. This distinguishes the impact of width from the effect of

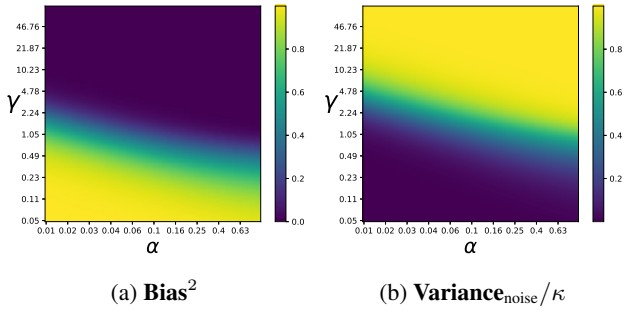

(a) **Bias**$^2$          (b) **Variance**$_{\text{noise}}/\kappa$

Figure 15: Expressions of **Bias**$^2$ and **Variance**$_{\text{noise}}/\kappa$ with $\tau = \gamma$.

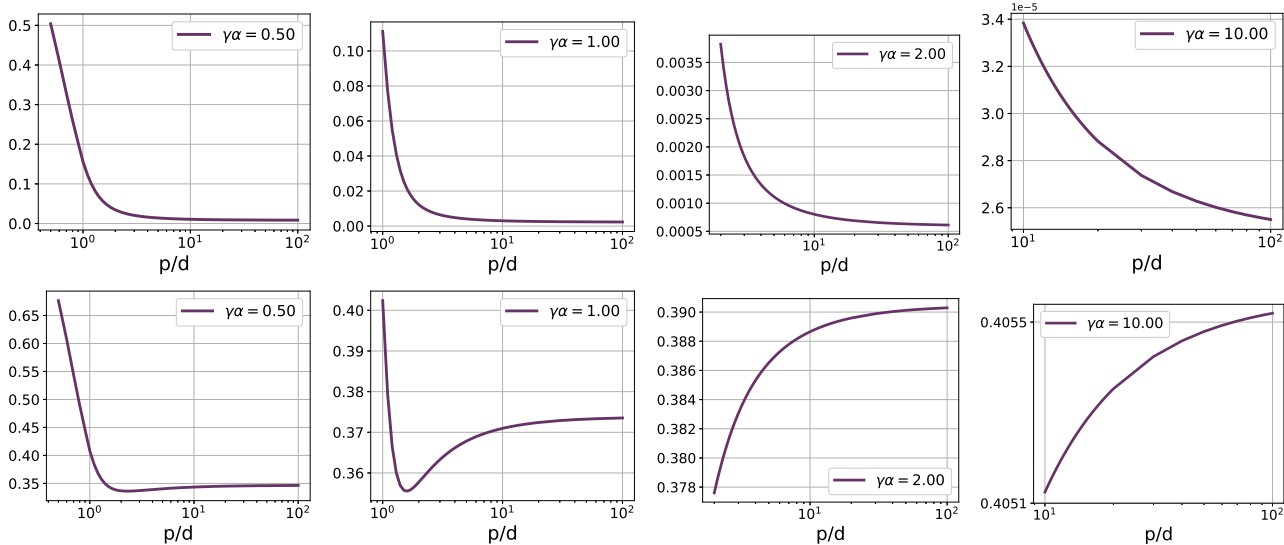

Figure 16: Risk curve with fixed $\gamma\alpha$. **Top:** $\kappa = 0$, the test loss decreases with width. **Bottom:** $\kappa = 0.64$, the test is either U-shaped or increasing.

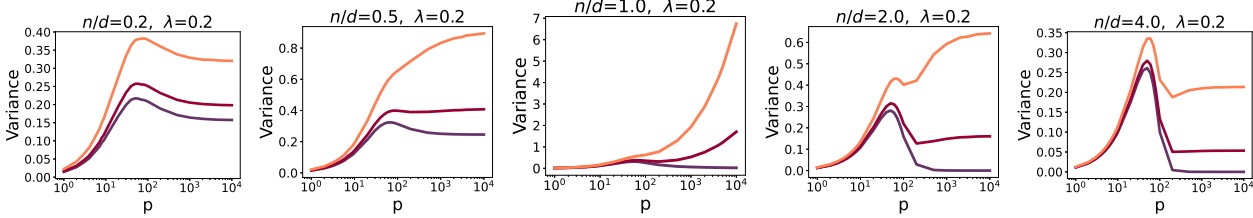

Figure 17: We plot the total variance in random feature ridge regression while fixing $d = 100$ and $\lambda = 0.2$, and varying $\sigma^2$. Legends show the values of $\sigma^2$, and titles show the values of $n/d$.

increasing model capacity. In our experiments, we fix $\gamma\alpha$ and plot the risk curve with varying $\gamma$ (subject to $\alpha \leq 1$) in Figure 16. For $\kappa = 0$, the test loss decreases with width. However, in the presence of large noise, the curve's shape can be altered. For $\kappa = 0.64$, the test loss exhibits either a U-shaped curve or an increasing trend.

# B    ADDITIONAL RESULTS FOR RANDOM FEATURE RIDGE REGRESSION WITH DIFFERENT $n/d$ RATIOS

Figure 17 shows the shape of total variance when $\lambda = 0.2$. Figures 18 and 19 show the shape of $\frac{1}{\sigma^2}$**Variance**$_{\text{noise}}$ with $\lambda = 0.2$ and $0.01$, respectively.

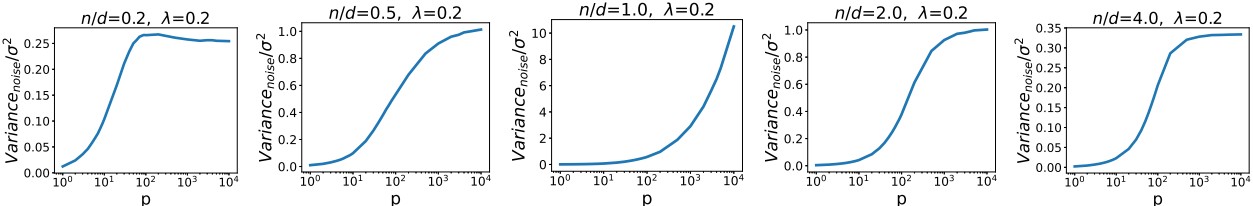

Figure 18: $\frac{1}{\sigma^2}\mathbf{Variance}_{\text{noise}}$ under different values of $n/d$ with $\lambda = 0.2$.

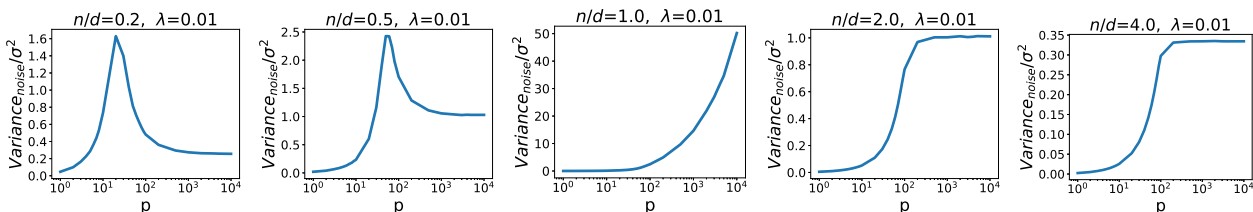

Figure 19: $\frac{1}{\sigma^2}\mathbf{Variance}_{\text{noise}}$ under different values of $n/d$ with $\lambda = 0.01$.

## C  EXPERIMENTAL SETTING DETAILS FOR NEURAL NETWORKS

All experiments are implemented using PyTorch. We use eight Nvidia A40 to run the experiments.

### C.1  NOISE TYPES

**Symmetric noise**   Symmetric noise is generated by randomly shuffling the labels of certain fraction of examples.

**Asymmetric noise**   Asymmetric noise is class-dependent. We follow the scheme proposed in Patrini et al. [2017] which is also widely used in robust method papers (e.g., ELR Liu et al. [2020] and DivideMix Li et al. [2020]). For CIFAR-10, labels are randomly flipped according to the following map: TRUCK→AUTOMOBILE, BIRD→AIRPLANE, DEER→HORSE, CAT→DOG, DOG→CAT. For CIFAR-100, since the 100 classes are grouped into 20 superclasses, e.g. AQUATIC MAMMALS contains BEAVER, DOLPHIN, OTTER, SEAL and WHALE, we flip labels of each class into the next one circularly within super-classes. For both datasets, the fraction of mislabeled examples in the training set is the noise level.

**Web Noise**   Red Stanford Car Jiang et al. [2020] contains images crawled from web, with label noise introduced through text-to-image and image-to-image search. There are 10 different noise levels {0%, 5% 10%, 15%, 20%, 30%, 40%, 60%, 80%} for this dataset and we choose 40% and 80%. For each noise level, the mislabeled web images can only be downloaded from the provided URLs. The training splits and labels are in provided files [1]. The original dataset size is 8144. For 80% noise, there are 6469 web images. However, 590 of the URL links are not functional and among the downloaded JPG files 1871 are corrupted/unopenable, hence we end up with 1675 clean examples and 4008 noisy examples, i.e., the actual noise level is 70.53%. For 40% noise, there are 3241 web images with 313 non-downloadable and 963 not unopenable. Therefore the actual noise level is 29.03%.

### C.2  EMPIRICALLY MEASURING BIAS AND VARIANCE

To empirically examine the transition in variance shape as suggested by Theorem 3.1, we adopt the unbiased estimator proposed in Yang et al. [2020]. Specifically, we randomly divide the training set for each model into $N$ subsets (with $N = 20$ for MNIST and $N = 5$ for CIFAR-10). For each subset, we train a separate model and compute the variance of the network output across these subsets. We use the mean squared error (MSE) loss on both datasets, as the bias-variance decomposition is only well-defined for MSE (refer to Eq. 1). While Yang et al. [2020] also proposed an estimator for cross-entropy loss, it is biased and may introduce skewed results.

---

[1]See their webpage for details `https://google.github.io/controlled-noisy-web-labels/download.html`

## C.3 TRAINING DETAILS FOR SECTIONS 4.2 AND 4.3

We see weight decay and $\ell_2$ regularization as equivalent terms. Thus, when we specify $\lambda = 0.001$, it indicates that we employ a weight decay value of 0.001. The width of a two-layer network is controlled by the number of hidden neurons. The width of a ResNet is controlled by the number of convolutional layer filters: for width $w$, there are $w$, $2w$, $4w$, $8w$ filters in each layer of the four Residual Blocks, respectively. When reducing the density of a model, we randomly select a certain fraction of its weights and then keep them zero throughout the training.

**MNIST and CIFAR-10 with MSE loss and Symmetric noise.** We train two-layer ReLU networks on MNIST and ResNet34 on CIFAR-10. On MNIST, we train each model for 200 epochs using SGD with batch size 64, momentum 0.9, initial learning rate 0.1, learning rate decay of 0.1 every 50 epochs. On CIFAR-10 we train each model for 1000 epochs using SGD with batch size 128, momentum 0.9, initial learning rate 0.1, learning rate decay of 0.1 every 400 epochs.

**CIFAR-10/100 with CE loss and Asymmetric/Symmetric noise** On both datasets, we train ResNet34 for 500 epochs using SGD with batch size 128, momentum 0.9, initial learning rate 0.1, learning rate decay of 0.1 every 100 epochs. We train ViT for 200 epochs using Adam with batch size 512, weight decay 0.001, and learning rate 0.0001.

**InceptionResNet-v2 on Red Stanford Car** We train each model for 160 epochs with an initial learning rate of 0.1, and a weight decay of $1 \times 10^{-5}$ using SGD and momentum of 0.9 with a batch size of 32. We anneal the learning rate by a factor of 10 at epochs 80 and 120, respectively. We use Cross Entropy loss.

## C.4 TRAINING DETAILS FOR SECTION 4.5

ELR leverages the early learning phenomenon where the network fits clean examples first and then mislabeled examples. It hinders learning wrong labels by regularizing the loss with a term that encourages the alignment between the model's prediction and the running average of the predictions in previous rounds. DivideMix dynamically discards labels that are highly likely to be noisy and trains the model in a semi-supervised manner. For both ELR and DivideMix we use the same setup as in the original papers Liu et al. [2020], Li et al. [2020].

**ELR** We train ResNet-34 using SGD with momentum 0.9, a weight decay of 0.001, and a batch size of 128. The network is trained for 120 epochs on CIFAR-10 and 150 epochs on CIFAR-100. The learning rate is 0.02 at initialization, and decayed by a factor of 100 at epochs 40 and 80 for CIFAR-10 and at epochs 80 and 120 for CIFAR-100. We use 0.7 for the temporal ensembling parameter, and 3 for the ELR regularization coefficient.

**DivideMix** We train ResNet-34 for 200 epochs using SGD with batch size 64, momentum 0.9, initial learning rate 0.02, a weight decay of $5 \times 10^{-4}$, and a learning rate decay of 0.1 at epoch 150. We use 150 for the unsupervised loss weight.

# D ADDITIONAL EXPERIMENTAL RESULTS FOR SECTION 4

## D.1 RESULTS FOR VIT

Figure 20 showcases the results of training ViT on CIFAR-10 across varying sample sizes. It is observed that ViT exhibits final ascent and the phenomenon is mitigated by increasing the sample size. These findings are consistent with observations in other scenarios discussed in Section 4.2.

## D.2 EFFECT OF WIDTH

**Additional results for Figures 6 and 7.** Test accuracy, **Bias**$^2$ and **Variance** are shown in Figures 21 to 24.

**Effect of Sample Size.** We use MSE loss on MNIST and CE loss on CIFAR-10. We consider $\lambda = 0.001$ and 60% noise for both datasets. Plots of test accuracy are shown in Figure 25.

## D.3 EFFECT OF DENSITY

**Joint effect of width and density** When studying the joint effect of width and density, we utilize MSE loss to measure bias and variance. We set $\lambda = 0$ for MNIST and $\lambda = 0.0005$ for CIFAR-10. The results are presented in Figures 11, 26, and 27.

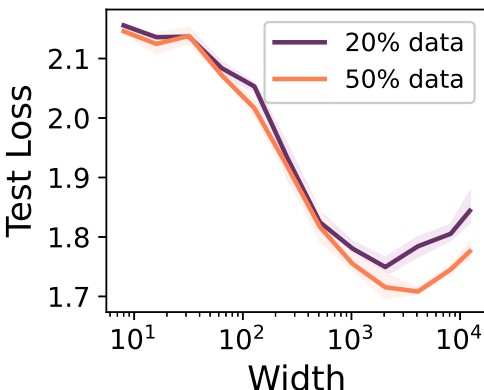

Figure 20: Test loss vs. width for ViT trained on CIFAR-10, with different sample sizes. We observe that the final ascent occurs for ViT, and that a larger sample size mitigates this final ascent. This aligns with our observations in other settings in Section 4.2.

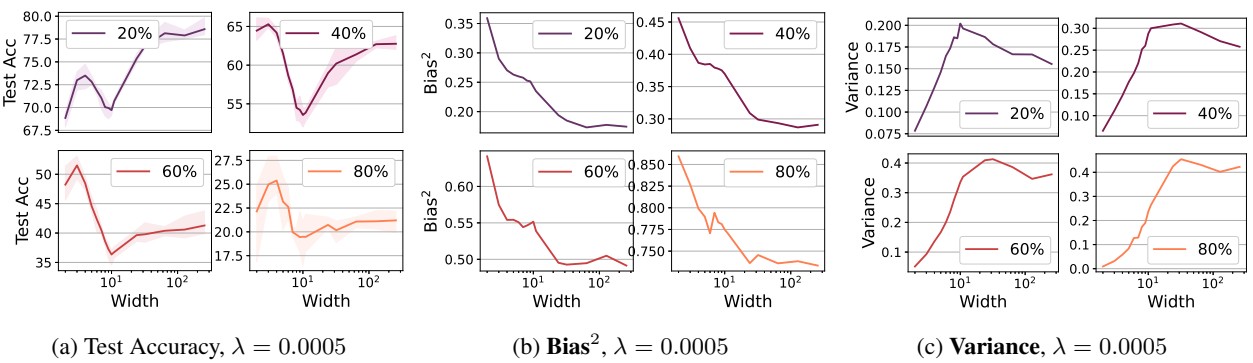

(a) Test Accuracy, $\lambda = 0.0005$     (b) **Bias**$^2$, $\lambda = 0.0005$     (c) **Variance**, $\lambda = 0.0005$

Figure 21: CIFAR-10, MSE loss, $\lambda = 0.0005$

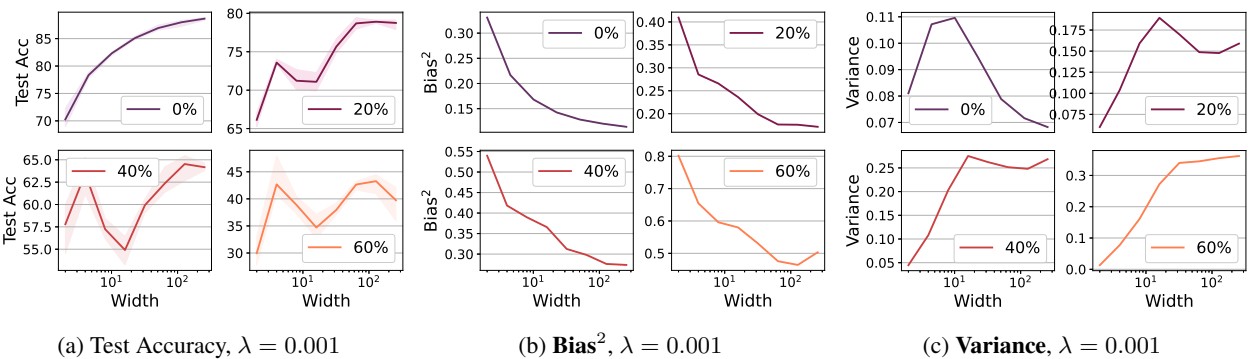

(a) Test Accuracy, $\lambda = 0.001$     (b) **Bias**$^2$, $\lambda = 0.001$     (c) **Variance**, $\lambda = 0.001$

Figure 22: CIFAR-10, MSE loss, $\lambda = 0.001$

In both settings, models with the smallest density achieve the highest accuracy, and models with very small density achieve the optimal loss.

**Smaller density vs. stronger $l_2$ regularization** We train ResNet34 with a width of 16 on CIFAR-10, using 50% symmetric noise. We compare the results obtained by varying $\lambda$ (weight decay) and by varying the model density. Figure 11d shows the test loss and 28 shows the test accuracy.

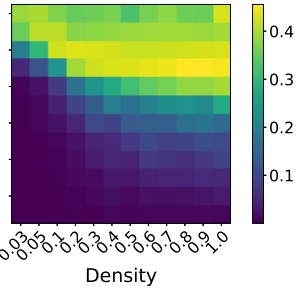

(a) Test Accuracy, $\lambda = 0$     (b) **Bias$^2$**, $\lambda = 0$     (c) **Variance**, $\lambda = 0$

Figure 23: MNIST, MSE loss, $\lambda = 0$

(a) Test Accuracy, $\lambda = 0.001$     (b) **Bias$^2$**, $\lambda = 0.001$     (c) **Variance**, $\lambda = 0.001$

Figure 24: MNIST, MSE loss, $\lambda = 0.001$

(a) MNIST with MSE loss, Test Accuracy     (b) CIFAR-10 with CE loss, Test Accuracy

Figure 25: We plot test accuracy against width while varying sample size. We use MSE loss with $\lambda = 0.001$ on MNIST and CE loss with $\lambda = 0.001$ on CIFAR-10.

Figure 26: **Variance** under varied width and density on CIFAR-10.

## D.4    WHEN ROBUST METHODS ARE APPLIED

In addition to the experiments presented in the main paper, we provide results for the following experiments conducted on CIFAR-10/100: width experiments with ELR under 80% noise (Figure 29), density experiments with ELR (Figure 30), density experiments with DivideMix under 60% noise (Figure 31). Furthermore, we show the results of density experiments

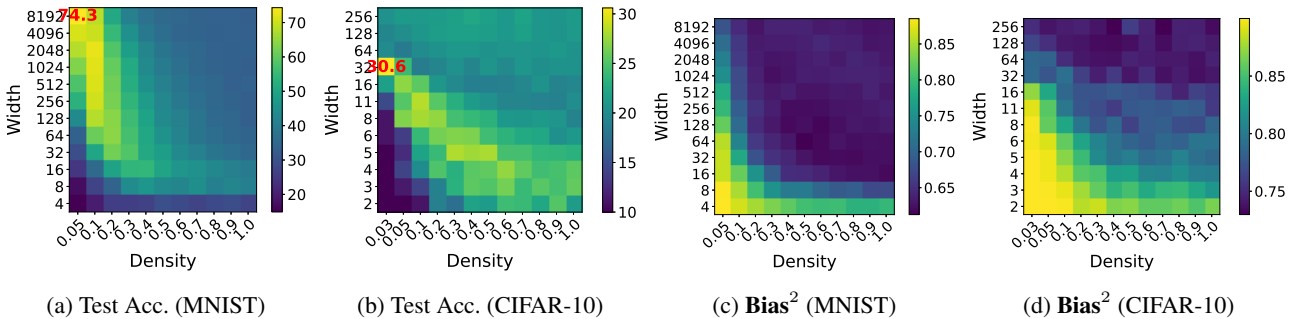

(a) Test Acc. (MNIST)  (b) Test Acc. (CIFAR-10)  (c) **Bias**$^2$ (MNIST)  (d) **Bias**$^2$ (CIFAR-10)

Figure 27: Test accuracy and bias under varied width and density. Red numbers show the highest accuracy.

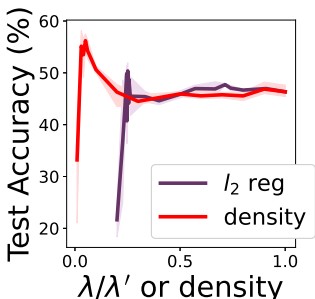

Figure 28: Test accuracy obtained by varying $\lambda$ (weight decay) or by varying model density.

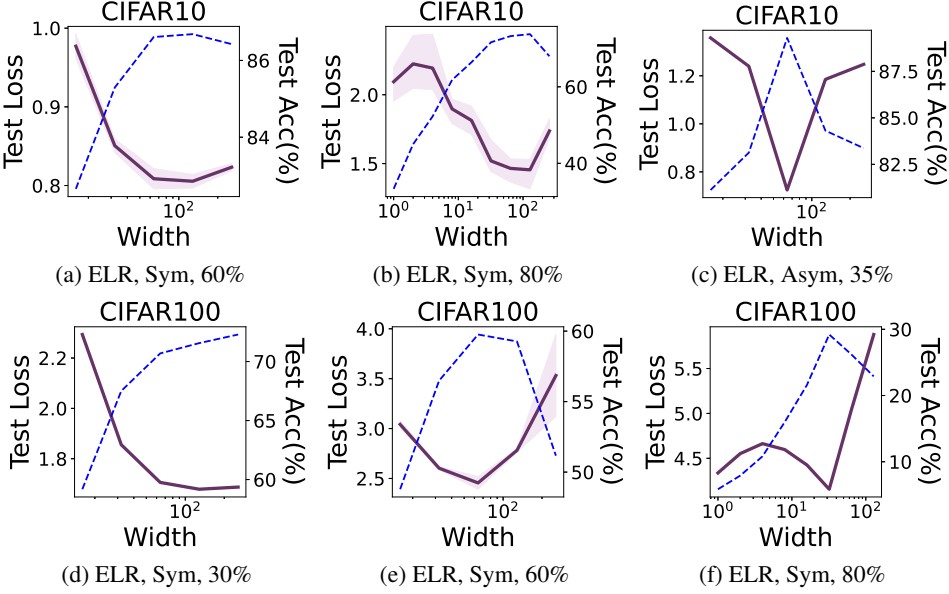

(a) ELR, Sym, 60%  (b) ELR, Sym, 80%  (c) ELR, Asym, 35%

(d) ELR, Sym, 30%  (e) ELR, Sym, 60%  (f) ELR, Sym, 80%

Figure 29: Additional results for the effect of width when ELR is used. The top row displays the results on CIFAR-10, and the bottom row displays the results on CIFAR-100.

on Red Stanford Car with 70% noise, where InceptionResNet-v2 is trained using ELR in Figure 32. In the plots, the purple solid line represents test loss and the blue dashed line represents test accuracy.

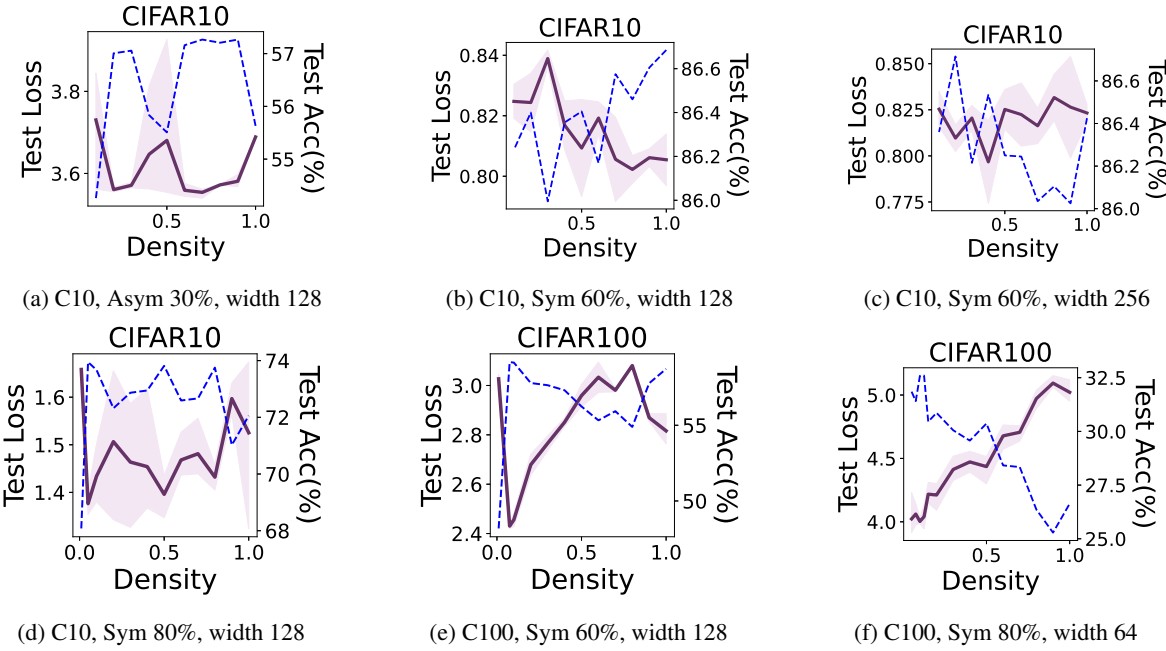

Figure 30: Effect of density when ELR is used.

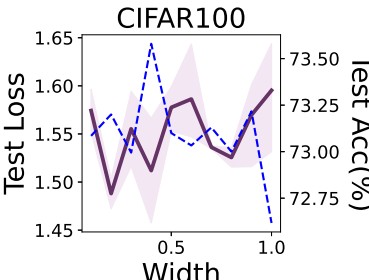

Figure 31: Effect of density when DivideMix is used. We train the model on CIFAR-100 and set width to 64.

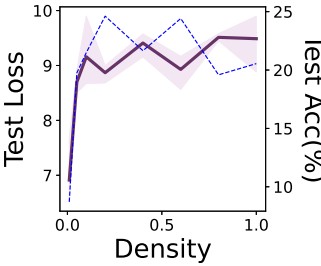

Figure 32: Effect of density when we train an InceptionResNetV2 on Red Stanford Cars using ELR.

# E INVESTIGATING THE COMPLEXITY OF LEARNED FUNCTIONS

A hypothesis proposed by Belkin et al. [2019] for double descent suggests that the learning algorithm possesses the right inductive bias towards "low complexity" functions that generalize well while fitting the training set. It posits that larger model sizes, which correspond to richer function classes, provide the algorithm with more choices to discover functions with lower complexity. Given our finding that large noise alters the correlation between model size and generalization, a natural question arises: "Does large noise also change the correlation between model size and the complexity of learned functions?" In this section, we present empirical evidence suggesting the possibility of an affirmative answer. Recently, Kalimeris et al. [2019] qualitatively measured the complexity of neural networks based on how much their predictions could be explained

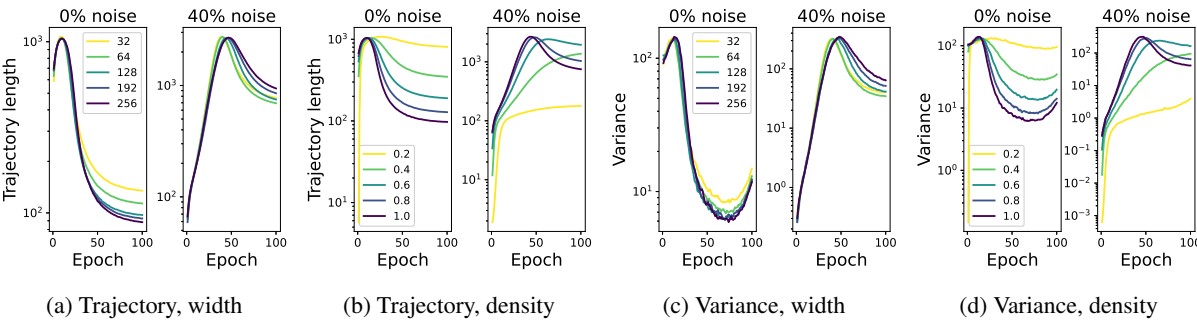

(a) Trajectory, width      (b) Trajectory, density      (c) Variance, width      (d) Variance, density

Figure 33: Trajectory length and variance of the first layer bias during each epoch. When varying the density we fix the width to 128.

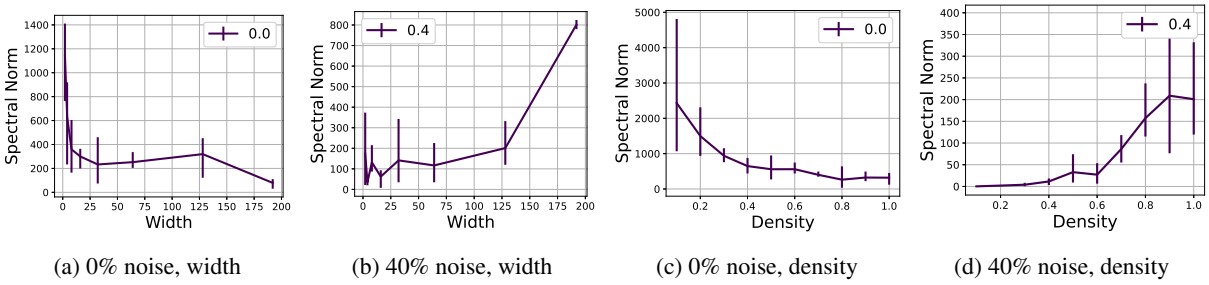

(a) 0% noise, width      (b) 40% noise, width      (c) 0% noise, density      (d) 40% noise, density

Figure 34: When width or density increases, spectral norm decreases under 0% noise but increases under 40% noise.

by a smaller model. However, this approach assumes that "smaller models have lower complexity," which may not hold in our case. Instead, we consider the following three measures:

- Trajectory length of the first layers bias. $\sum_{t \in T} \frac{\|\boldsymbol{b}_1^{(t+1)} - \boldsymbol{b}_1^t\|_2}{\alpha_t \epsilon_{f(t)}}$ where $T$ is a set of iteration indices, $\boldsymbol{b}_1^{(t)}$ is the parameter of the first layer bias at iteration $t$, and $\epsilon_{f(t)}$ is the gradient of the loss w.r.t. the network's output at epoch $t$. Loukas et al. [2021] shows that, under certain conditions, the above can be both upper and lower bounded in terms of the Lipschitz constants of functions represented by the network at iterations in $T$ (see their Theorem 1). This implies that the first layer bias travels longer during training when it is fitting a more complex function.

- Variance of the first layer bias $\mathrm{avg}_{t \in T}\|\boldsymbol{b}_1^{(t)} - \mathrm{avg}_{t \in T}\boldsymbol{b}_1^{(t)}\|_2^2$. Loukas et al. [2021] also provides a lower bound for the above in terms of the Lipschitz constant and $\epsilon_{f(t)}$ (their Corollary 2). Thus a larger Lipschitz constant leads to a higher lower bound, meaning that when fitting a lower complexity function, the network's bias will update more frequently during training.

- Product of spectral norms of the layer parameters. This is known as an upper bound for the network's Lipschitz constant Szegedy et al. [2013]. For convolutional layers, the spectral norm is computed using the FFT-based algorithm in Sedghi et al. [2018].

Our experimental setup is the same as that of Loukas et al. [2021] (Task 2 in Section 6). We train CNNs on CIFAR-10 DOG vs AIRPLANE. The CNN consists of one identity layer, two convolutional layers with a kernel size of 5, a fully connected ReLU layer with a size of 384, and a linear layer. The width of the CNN is controlled by the number of convolutional channels, with $16w$ channels in each convolutional layer for width $w$. We employ binary cross-entropy (BCE) loss and train the models for 200 epochs using vanilla SGD with a batch size of 1. We apply exponential learning rate decay with a factor of $10^{-50}$. The trajectory length and variance are computed at every epoch, reflecting the complexity of the CNN's represented function at each stage. The results are depicted in Figure 33, showing that noise can alter the relative complexity of functions learned by models with increasing width or density. This trend is more pronounced in the case of width (Figures 33a and 33c), where both quantities decrease under 0% noise and increase under 40% noise as width increases. Figure 34 presents the product of layer-wise spectral norms of the neural network at the final epoch, showing a similar pattern. These results suggests that label noise can invert the originally negative correlation between size and complexity/smoothness.

# F    POTENTIAL CONNECTION TO BENIGN/CATASTROPHIC OVERFITTING

The term 'benign overfitting' describes the phenomenon where models trained to overfit the training set still achieve nearly optimal generalization performance Bartlett et al. [2020], Tsigler and Bartlett [2020], Chatterji et al. [2021], Cao et al. [2022], Frei et al. [2022], Mallinar et al. [2022]. Recently, Cao et al. [2022] demonstrated that sufficiently large models exhibit benign overfitting when the product of sample size and signal-to-noise ratio (SNR) is large, and catastrophic overfitting occurs otherwise. Notably, this condition coincides with the condition in our theory (Section 3) where the final ascent does not occur, since the SNR is represented as $1/\sigma^2$ in our setting. Consequently, our findings can be interpreted in the context of benign/catastrophic overfitting: when the noise-to-sample-size ratio is large, a 'sufficiently large' model overfits catastrophically, implying that increasing the model size towards a sufficient extent may worsen generalization. Indeed, neural networks and various other real interpolating methods typically operate in the 'tempered overfitting' regime Mallinar et al. [2022]. Our results suggest that the condition for benign overfitting identified in Cao et al. [2022] can potentially be extended to assess the relative 'benignity' of tempered overfitting across different model sizes. The theoretical establishment of this connection could be a valuable direction for future research.