# OpenReview forum: "Investigating the Impact of Model Width and Density on Generalization in Presence of Label Noise"
_auai.org/UAI/2024/Conference — UAI 2024 spotlight_

### Official Review · Reviewer_P2ta · 2024-02-26

**Q2-1 Originality-Novelty:** 3
**Q2-2 Correctness-Technical Quality:** 3
**Q2-5 Clarity Of Writing:** 3

**Q10 Ethical Concerns:**

None.

**Q1 Summary And Contributions:**

The research explores the impact of label noise on the well-known double descent phenomenon in overparameterized neural networks. While increasing network size is vital for achieving state-of-the-art performance, the study reveals a novel observation: label noise introduces a final ascent in the double descent curve, with optimal generalization occurring at intermediate widths under a sufficiently large noise-to-sample-size ratio. The theoretical analysis attributes this to the shape transition of test loss variance induced by label noise. The study extends this final ascent phenomenon to model density, showing that reducing density by randomly dropping trainable parameters enhances generalization under label noise. This paper also shows that larger $\ell_2$ regularization and robust learning methods exacerbate the final ascent. Experimental validation is conducted on various architectures, confirming these findings.

**Q2-3 Extent To Which Claims Are Supported By Evidence:**

3: Good: the main claims are supported by convincing evidence (in the form of adequate experimental evaluation, proofs, (pseudo-)code, references, assumptions).

**Q2-4 Reproducibility:**

3: Good: key resources (e.g. proofs, code, data) are available and key details (e.g. proofs, experimental setup) are sufficiently well-described for competent researchers to confidently reproduce the main results.

**Q3 Main Strengths:**

- The motivation of this paper is strong and clear.
- The technical contributions are great.
- Experiments overall support the claims well.

**Q4 Main Weakness:**

- The writing can be polished to improve legibility.
- Experiments can be more convincing.

**Q5 Detailed Comments To The Authors:**

- This paper claims lower density for wider models allows for improved generalization under label noise. This claim is a bit confusing for me. First, there may be a trade-off between density and generalization. If the density is too low, the model may under-fit the training data, which hurts generalization. Second, if the model is wide, the fitting power will be improved. If so, the model may overfit mislabeled data. Therefore, the claim is a bit confusing.
- Does the claim be true under other methods, not just regularization, e.g., loss correction [1]?
- The claim about density is related to [2], which shows that if we control the fraction of trainable weights, the generalization can be enhanced. More discussions are needed.
- For Section 3.1, the formulation is different from the traditional label noise problem, since the label noise seems to be continuous rather than discrete.
- Did the paper try other forms of label noise concerning the above problem?
----
[1] Are anchor points really indispensable in label-noise learning? In NeurIPS 2019.
[2] Robust early-learning: Hindering the memorization of noisy labels. In ICLR 2021.

**Q9 Complying With Reviewing Instructions:**

Yes

---

> ### Author Rebuttal · Authors · 2024-04-03
>
> We thank the reviewer for recognizing the paper's strong motivation, significant technical contributions, and well-supported claims through experiments. Below we reply to the comments.
>
> > regarding wider but sparser models
>
> Our paper discusses two distinct cases, for which the insights are clearly discussed in sections, 3.2 and 3.3, and in the experimental sections, 4.3 and 4.4. We believe there might be a confusion of the two settings. Here, we will reiterate to further clarify.
> 1. Varying density while keeping width fixed: In this scenario, the trade-off mentioned by the reviewer holds true. With sufficient noise, generalization improves and then worsens as density increases, as we have extensively shown and discussed in Sec 3.2, Fig 4b, Sec 4.3, and Fig 10.
> 2. Varying both density and width. The claim that ”lower density for wider models allows for improved generalization under label noise” pertains to scenarios where both density and width are varied **simultaneously**. In such cases, generalization **monotonically** improves by *increasing the width while further reducing the density*, as demonstrated in Fig 4d and Figs 11a, 11b. In the **overparameterized regime** , which is beyond what can be explained by the simple underfitting-overfitting narrative mentioned by the reviewer, a more plausible explanation is: with wider but sparser models, the entire function space (all functions that can be represented by the model) contains more ‘simple’ minimizers of the training loss. Consequently, the minimizer of the training loss tends to be simpler and, intuitively, generalizes better. We note that providing a precise explanation for this intriguing phenomenon represents a future research direction and may require significant effort from researchers in this field to fully unravel.
>
> > whether the claim is true for loss correction method
>
> We will include the following discussion and the referenced paper in our final version. In general, all robust methods can be thought of as a form of implicit regularization at a very high level, since they share the common intuition of preventing the model from fitting certain data too closely. For example, ELR boosts the gradient of examples with likely clean labels and neutralizes the gradient of examples with likely incorrect labels. Similarly, DivideMix discards the labels of data that are likely incorrect, thus preventing the model from fitting to noisy labels. From this perspective, loss correction methods can also be considered in this way, as it effectively puts more weights on the clean data, making the model fit the noisy data less. Therefore, we believe the same conclusion would likely apply to other methods as well.
>
> > The claim about density is related to [2], which shows that if we control the fraction of trainable weights, the generalization can be enhanced. More discussions are needed.
>
> We will include the discussion in the final version. [2] divides the model weights into two sets, updating one of them while shrinking the other, with the division dynamically adjusted during training to combat label noise. This demonstrates that learning fewer weights can, in some sense, benefit robustness. Our findings make an even stronger statement: even by simply removing some weights randomly from the very beginning of training, robustness can be enhanced with an appropriate weight removal ratio. The implication of our research is that naively dropping weights is a baseline worth considering in this context. Additionally, we provide theoretical groundwork for understanding the effect of methods similar to [2], showing that much of the benefit can be viewed as simply reducing the hypothesis space. From the bias-variance perspective, reducing density can lead to a smaller noise-dependent variance. We will include the above discussion in the revised version.
>
> > or Section 3.1, the formulation is different from the traditional label noise problem, since the label noise seems to be continuous rather than discrete.
>
> This is because our analysis is on a regression task instead of classification, and uses the standard way of considering label noise in regression. Since the labels in regression are not categorical, it does not make sense to have discrete noise. Regarding the relevance of analyzing regression and how this helps us understand classification, we have discussed this at the beginning of Section 3. Essentially, this approach follows the rich prior literature on characterizing the generalization curve and is, so far, the only theoretically feasible way of addressing this problem. Furthermore, analyzing regression is known to have played a pivotal role in providing significant insights into understanding generalization in classification with real neural networks.

---

### Official Review · Reviewer_anWz · 2024-02-27

**Q2-1 Originality-Novelty:** 3
**Q2-2 Correctness-Technical Quality:** 3
**Q2-5 Clarity Of Writing:** 2

**Q1 Summary And Contributions:**

1. Larger model sizes impair generalization in the presence of label noise.
2. Lower density for wider models allows for improved generalization under label noise.

**Q2-3 Extent To Which Claims Are Supported By Evidence:**

3: Good: the main claims are supported by convincing evidence (in the form of adequate experimental evaluation, proofs, (pseudo-)code, references, assumptions).

**Q2-4 Reproducibility:**

3: Good: key resources (e.g. proofs, code, data) are available and key details (e.g. proofs, experimental setup) are sufficiently well-described for competent researchers to confidently reproduce the main results.

**Q3 Main Strengths:**

See Comments.

**Q4 Main Weakness:**

See Comments.

**Q5 Detailed Comments To The Authors:**

I believe this paper essentially meets the acceptance criteria for UAI.

The motivation of the paper is strong enough, and I agree with the author's view that the effect of label noise on the double descent phenomenon has not been fully explored. Although this paper may not provide new theories and techniques, and the phenomena discovered are not surprising, the author's detailed explanations of these phenomena provide valuable insights into the roles of regularization in learning with noisy labels.

The related work in the paper is quite solid. It is worth mentioning that the deep double descent is not only width-wise but also epoch-wise. Enhancing the understanding of epoch-wise learning with label noise (for examples, [1, 2, 3]) might help the author further explain the phenomenon of the final ascent in width-wise discussions.

The paper may have the following weaknesses:

1. The main conclusions of this paper are confusing. The finding of final ascent, that larger model sizes impair generalization in the presence of label noise, is intuitive, especially considering the high noise rate in the experimental settings of this paper. At the same time, it lacks new insights that could guide applications ("Lower density for wider models allows for improved generalization under label noise" might be useful, but lacks empirical studies in specific applications). I hope the authors can clarify this point. This weakness may be related to unclear writing and layout of the paper.
2. I have also observed that models with intermediate width or density can be even more beneficial when use some robust methods, the author's attempt to elucidate it is commendable. However, the perspective presented in the related work section—that reducing the model width or density can yield even greater benefits when training with robust methods—may not be entirely accurate. Not all robust methods can be simplistically equated to the assumption of stronger L2 regularization. In such instances, these robust methods rely on the enhanced learning capabilities of width or density networks to improve model robustness. Exploring a wider array of regularization methods could offer substantial benefits to the paper's contributions.

It is worth mentioning that, despite the above-mentioned weaknesses, the overall quality of this paper is satisfactory.

- [1.] Phases of learning dynamics in artificial neural networks in the absence or presence of mislabeled data. Machine Learning: Science and Technology, 2021.
- [2.] Characterizing Datapoints via Second-Split Forgetting. NeurIPS 2022.
- [3.] Early Stopping Against Label Noise Without Validation Data. ICLR 2024.

**Q9 Complying With Reviewing Instructions:**

Yes

---

> ### Author Rebuttal · Authors · 2024-04-03
>
> We thank the reviewer for acknowledging the strong motivation of our paper, and for appreciating our detailed explanation of new phenomena and valuable insights into the roles of regularization. Below we reply to the comments.
>
> > … Although this paper may not provide new theories and techniques, and the phenomena discovered are not surprising …
>
> Our paper indeed provides new theories, including Thm 3.1, which presents the first analytical result from which one can observe final ascent, and *in particular Thm 3.2, which is the first theory that extends the discussion of generalization behavior to model density*, thereby uncovering final ascent with respect to model density.
>
> We also believe that the phenomena we revealed are not necessarily expected, in particular items 2 and 3 below.
> 1. the transition of variance shape caused by label noise, which shows a more complex picture than what shown in prior work, where the variance can evolve beyond unimodal to exhibit an increasing-decreasing-increasing pattern;
> 2. that wider but sparser models, despite having more parameters, still achieves monotonically improving generalization when scaling;
> 3. that stronger regularization/robust methods exacerbate final ascent, which was not anticipated, as they were expected to counteract label noise.
>
>
> > epoch-wise double descent/final ascent
>
> Thank you for the valuable suggestion. While this work primarily focuses on model-size-wise final ascent, we do agree that investigating whether an epoch-wise final ascent occurs and understanding the underlying mechanism could be a valuable direction for future research. We hope our work can serve as the first step and inspire further investigation in this area. We will add the discussion to our revised version.
>
>
> > Main conclusions of the paper
>
> Here, we reiterate the main conclusions, which have been mentioned in the latter part of the introduction and Sec 5:
> - Final ascent occurs with respect to both model width and density.
> - Transition in variance shape caused by label noise.
> - Stronger L2 regularization or robust methods exacerbate final ascent.
> - Benefits of wider but sparser networks.
> - Larger sample sizes alleviate final ascent.
>
> > .. it lacks new insights that could guide applications ("Lower density for wider models allows for improved generalization under label noise" might be useful, but lacks empirical studies in specific applications) ..
>
> To highlight the **practical application of reducing model density**, we have studied several practical scenarios:
> 1. In Figs 30 and 32, we demonstrated that *reducing model density further improves the performance of SOTA robust learning algorithms*.
> 2. Additionally, Figs 11a and 11b showcase the *benefits of wider but sparser models on real datasets*. The above experiments were conducted on MNIST using MLPs and on CIFAR-10 using ResNets, representing real-world practical scenarios. Please note that large scale experiments require very large computational resources that are not available to us (e.g., training ResNet34-Width256 on Cifar10 requires 20GB GPU memory, and plotting just the first row of Fig 11 b,c requires training 60 such models; training ResNet34-Width512 would require 45.5GB memory which exceeds the capacity of our GPU).
> 3. Another practical insight is the importance of using intermediate widths and densities in settings with limited data, which is particularly relevant in scenarios where data is scarce, such as in the medical field. While our work studied standard vision benchmarks, we expect the findings of our paper to benefit many different applications in the future.
>
> We will include the above discussion in our revision.
>
> > Not all robust methods can be simplistically equated to the assumption of stronger L2 regularization …
>
> While we agree that robust methods are not exactly equivalent to l2 regularization, at a very high level they can generally be thought of as a form of implicit regularization, since they share the common intuition of preventing the model from fitting certain data too closely. For example, ELR boosts the gradient of examples with likely clean labels and neutralizes the gradient of examples with likely incorrect labels. Similarly, DivideMix discards the labels of data that are likely incorrect, thus preventing the model from fitting to noisy labels.
> We have theoretically shown that L2 regularization exacerbates final ascent. By making such a connection, we seek a potential explanation for our observation that robust methods exacerbate the final ascent. Therefore, we believe the same conclusion could likely apply to general robust methods as well.
>
> While we have confirmed that two SOTA robust methods, namely DivideMix and ERL, exacerbate the final ascent wrt both width and density (which already speak volumes and are of practical significance), we do agree a comprehensive empirical study on various robust methods is a valuable direction for future work.

---

### Official Review · Reviewer_ub8k · 2024-03-20

**Q2-1 Originality-Novelty:** 2
**Q2-2 Correctness-Technical Quality:** 3
**Q2-5 Clarity Of Writing:** 3

**Q1 Summary And Contributions:**

This paper investigates the effect of label noise on the test loss curve. While the test loss curve follows a "descent-ascent-descent" pattern without label noise, the authors first reported that label noise may cause a final ascent, which means that it is important to carefully set the scale of neural networks in the presence of label noise. Besides, the authors also introduced the notion of model density and also analyzed its effect on test loss. In addition, they also discussed the roles of regularization and sample size.

**Q2-3 Extent To Which Claims Are Supported By Evidence:**

3: Good: the main claims are supported by convincing evidence (in the form of adequate experimental evaluation, proofs, (pseudo-)code, references, assumptions).

**Q2-4 Reproducibility:**

3: Good: key resources (e.g. proofs, code, data) are available and key details (e.g. proofs, experimental setup) are sufficiently well-described for competent researchers to confidently reproduce the main results.

**Q3 Main Strengths:**

1. This paper provides comprehensive analyses. It not only reports that label noise causes a final ascent of the test loss curve but also analyzes the roles of model density, regularization, sample size, and robust learning algorithms.

2. This paper provides extensive experimental results. They train various networks on many benchmark datasets, which sufficiently validate their argument.

3. This paper is well-organized and well-written. By the way, the figures is pretty good.

**Q4 Main Weakness:**

1. Most theoretical results can be regarded as a minor extension of [1]. Although [1] didn't consider the influence of label noise while this paper does, I do not think there is a great technical gap between the two.

2. The theoretical results in this paper are based on a toy model, a linear neural network with a random first layer, I wonder whether the theoretical results can generalize to more complex models.

3. The final ascent is only evident in the presence of serious label noise (>50%), but in many real-world applications, the noise rate is typically less than 20%. For instance, Clothing 1M contains about 8% label noise while WebVision contains about 20% label noise. Therefore, I'm not sure whether the main argument in this paper is significant.


[1] Zitong Yang, Yaodong Yu, Chong You, Jacob Steinhardt, and Yi Ma. Rethinking bias-variance trade-off for generalization of neural networks. In International Conference on Machine Learning, pp. 10767–10777. PMLR, 2020.

**Q5 Detailed Comments To The Authors:**

Overall, I think this paper is a self-contained work. The authors argue that label noise introduces an additional ascent of test loss curve, and also analyze the roles of model density, regularization, sample size, and robust learning algorithms both theoretically and empirically. In particular, the experimental results are extensive and convincing. Besides, the presentation of this paper is good, making it easy to follow. Therefore, I think this paper deserves readership.

However, my main concern lies in the theoretical results. I think the authors only extend previous works slightly and the theoretical results are based on a toy model. I highly recommend the authors to generalize their theoretical results to more complex models, which can remarkably distinguish their work from previous ones. Therefore, I cannot further improve my score temporarily.

**Q9 Complying With Reviewing Instructions:**

Yes

---

> ### Author Rebuttal · Authors · 2024-04-03
>
> We thank the reviewer for appreciating the paper's comprehensive analyses, extensive and convincing experimental results, and acknowledging our thorough examination of various factors. Below we reply to the comments.
>
> > Most theoretical results can be regarded as a minor extension of [1]. Although [1] didn't consider the influence of label noise while this paper does, I do not think there is a great technical gap between the two.
>
> - Thm 3.1 on label noise indeed builds upon the analysis of [1], but it's important to note that the phenomenon of final ascent is completely novel and *cannot be observed from the expression derived in [1]*.
> - We also wish to highlight our novel contributions regarding model density. *We are the first to systematically explore the generalization behavior with respect to model density* (Thm 3.2) and uncover the final ascent phenomenon related to it.
> - Additionally, we believe that the value of our contribution is not solely determined by technical complexity but more significantly by the insights we uncover. These include the effects of regularization, robust methods, and sample size on the shape of the loss curve, which have not been studied before. The phenomenon of final ascent that we have identified is entirely novel and distinct from prior work.
>
> > “…I wonder whether the theoretical results can generalize to more complex models ..."
>
> Generalizing the analysis to nonlinear models is challenging, considering that the literature on the theoretical study of the generalization curve has not actually extended beyond linear models. However, one possible future step that we hope will see some progress in the near future is to analyze final ascent for linear models trained on nonlinear features (i.e., with a fixed random first layer having nonlinear activation after it). The furthest advancement we have observed is by Adlam in 2022. However, they did not have an analytical expression for non-zero regularization, which we expect to be necessary for observing final ascent. We hope this technical challenge can be addressed in the future.
>
>
> > “The final ascent is only evident in the presence of serious label noise (>50%), but in many real-world applications, the noise rate is typically less than 20%. For instance, Clothing 1M contains about 8% label noise while WebVision contains about 20% label noise. Therefore, I'm not sure whether the main argument in this paper is significant.”
>
> 1. Please note that the required noise level can be as low as 20-30% in many cases (e.g., Figures 6, 7, 8). As mentioned in the last paragraph of Section 4.2 (discussions), it is evident that the required noise level depends on several factors and is particularly lower when there is large regularization or a small sample size, or when the noise is more skewed (e.g., asymmetric). In practice, since realistic noise is more likely to be instance/class-dependent, thus asymmetric, and robust methods are necessarily used in the face of noise, we expect the required noise level to observe final ascent to be relatively low in realistic data.
> 2. Very high noise levels (50%-80%) are commonly considered in papers studying robust algorithms, e.g., (Zhang and Sabuncu 2018, Jiang et al. 2018, Han et al. 2018, Mirzasoleiman et al. 2020, Liu et al. 2020, Li et al. 2020). Since in practice the noise is often instant/class-dependent and asymmetric, methods are evaluated against a larger noise level. Hence there is a substantial number of papers addressing high noise, and we believe this setting is both highly relevant and of practical significance.

---

### Official Review · Reviewer_j8kX · 2024-03-23

**Q2-1 Originality-Novelty:** 3
**Q2-2 Correctness-Technical Quality:** 3
**Q2-5 Clarity Of Writing:** 4

**Q10 Ethical Concerns:**

No.

**Q1 Summary And Contributions:**

This study investigates the generalization theory and empirical performance of deep learning models in the presence of label noise under different model widths and densities. This study reveals a new phenomenon named final ascent with a large noise-to-sample ratio when over-parameterized models share an increase in test loss w.r.t. increase in model width. Theoretical analysis revealed the role of variance induced by label noise and further investigated how model density affects generalization. Empirical studies confirmed the existence of the phenomenon and provided multiple insights.

**Q2-3 Extent To Which Claims Are Supported By Evidence:**

4: Excellent: all claims are supported by very convincing evidence (in the form of comprehensive experimental evaluation, rigorous mathematical proofs, detailed (pseudo-)code, precise references, well-motivated and realistic assumptions) and the authors deliver what they promise.

**Q2-4 Reproducibility:**

3: Good: key resources (e.g. proofs, code, data) are available and key details (e.g. proofs, experimental setup) are sufficiently well-described for competent researchers to confidently reproduce the main results.

**Q3 Main Strengths:**

1. Technically solid paper with both theoretical analysis and empirical studies on an ignored topic.
2. Claims are supported by convincing evidence and a complete picture has been established.
3. Contribution with insights and potential bridging effect in the study's field.

**Q4 Main Weakness:**

1. The study's novelty could be questioned by stating some label-to-noise ratios are not prevalent in deep learning practices.

**Q5 Detailed Comments To The Authors:**

1. The definition of some symbols could be emphasized in Figure captions for a quicker understanding.
2. A question I would like to discuss is if a high label-to-noise ratio is prevalent in modern deep learning practices. While the final ascent is robust across the experiment setups in this study, would a noise ratio > 0.4 too high in practice? Discussion and solid references would be well appreciated.
3. (minor)* Discussion: While presenting test loss curves under different label-to-noise ratios is much demonstrable, is it possible to raise a new measure of the full picture of generalization performance (U-shape, double descent, double plateau, final ascent)?

**Q9 Complying With Reviewing Instructions:**

Yes

---

> ### Author Rebuttal · Authors · 2024-04-03
>
> We thank the reviewer for recognizing our thorough exploration of an overlooked topic, our technically solid analysis, and the strong evidence supporting our claims. We appreciate the acknowledgment of our contribution's insights and its potential impact in the field. Below we reply to the comments.
>
> > some label-to-noise ratios are not prevalent
>
> 1. Please note that the required noise level can be as low as 20-30% in many cases (e.g., Figures 6, 7, 8). As mentioned in the last paragraph of Section 4.2 (discussions), it is evident that the required noise level depends on several factors and is particularly lower when there is large regularization or a small sample size, or when the noise is more skewed (e.g., asymmetric). In practice, since realistic noise is more likely to be instance/class-dependent, thus asymmetric, and robust methods are necessarily used in the face of noise, we expect the required noise level to observe final ascent to be relatively low in realistic data.
> 2. Very high noise levels (50%-80%) are commonly considered in papers studying robust algorithms, e.g., (Zhang and Sabuncu 2018, Jiang et al. 2018, Han et al. 2018, Mirzasoleiman et al. 2020, Liu et al. 2020, Li et al. 2020). Since in practice the noise is often instant/class-dependent and asymmetric, methods are evaluated against a larger noise level. Hence there is a substantial number of papers addressing high noise, and we believe this setting is both highly relevant and of practical significance.
>
> > Discussion: While presenting test loss curves under different label-to-noise ratios is much demonstrable, is it possible to raise a new measure of the full picture of generalization performance (U-shape, double descent, double plateau, final ascent)?
>
> Finding 'quantitative measures of the generalization behavior', is indeed an interesting research direction. Currently, generalization behavior (U-shape, double descent, double plateau, final ascent) is classified based on the shape of the loss curve, which requires visual inspection of a plot. The entire picture can potentially be represented by a numerical value. Although this poses a significant challenge, the foresight we can provide is that the measure should be a function of many factors, such as dataset, sample size, noise level, effective regularization (which depends on both explicit regularization and the training algorithm), and model architecture. However, both defining such a measure and devising practical ways to calculate it would be very challenging and require significant effort from researchers in this field.
>
> > The definition of some symbols could be emphasized in Figure captions for a quicker understanding
>
> Thank you for the suggestion. We will update the captions in the final version.

---

### Meta-Review · Area_Chair_7tDk · 2024-04-15

This paper investigates the impact of label noise on the double descent phenomenon in overparametrized deep networks, revealing that a final ascent phase occurs in certain cases. Specifically, the study demonstrates that as models increase in width, coupled with robust training techniques and a relatively high signal-to-noise ratio, sparse weights are required to maintain generalization. The paper is well-written, with clear motivation, solid technical content, and thorough theoretical analysis. Experimental results support the claims made. While there are concerns about the novelty of the approach and the simplified model setting considered, I believe that the contribution is still interesting.